# Variational Annealing on Graphs for Combinatorial Optimization

**Sebastian Sanokowski**[1,2]      **Wilhelm Berghammer**[2]      **Sepp Hochreiter**[1,2]

**Sebastian Lehner**[1,2]

[1] ELLIS Unit Linz and LIT AI Lab
[2] Institute for Machine Learning, Johannes Kepler University Linz, Austria
sanokowski@ml.jku.at

## Abstract

Several recent unsupervised learning methods use probabilistic approaches to solve combinatorial optimization (CO) problems based on the assumption of statistically independent solution variables. We demonstrate that this assumption imposes performance limitations in particular on difficult problem instances. Our results corroborate that an autoregressive approach which captures statistical dependencies among solution variables yields superior performance on many popular CO problems. We introduce subgraph tokenization in which the configuration of a set of solution variables is represented by a single token. This tokenization technique alleviates the drawback of the long sequential sampling procedure which is inherent to autoregressive methods without sacrificing expressivity. Importantly, we theoretically motivate an annealed entropy regularization and show empirically that it is essential for efficient and stable learning. [1]

## 1 Introduction

Combinatorial optimization (CO) problems are of central interest to a wide range of fields, including operations research, physics, and computational complexity [Papadimitriou and Steiglitz, 1998]. Since CO problems are typically NP-hard one would not expect that one can find the solutions to all arbitrarily large CO problem instances in polynomial time. However, these restrictions are concerned with worst case scenarios over entire problem families. For instance finding the Minimum Independent Set of *any* graph. Consequently, it is not surprising that a large body of work is focused on the design of solution strategies that yield a particularly good performance on a restricted subset of all possible instances. There is growing interest in exploring deep learning approaches to this restricted CO setting [Bello et al., 2017]. These methods aim to learn how to efficiently generate high quality solutions for CO problems and rely primarily on learned algorithmic components rather than on problem-specific, hand-crafted ones [Bengio et al., 2021]. Importantly, practicable methods should not rely on supervised training with solutions from other solvers since this limits the achievable performance of the learned model to that of the solver used to generate the training data [Yehuda et al., 2020]. As a result the development of unsupervised methods [Karalias and Loukas, 2020, Wang et al., 2022, Qiu et al., 2022, Karalias et al., 2022, Min et al., 2022, Wang and Li, 2023] and of Reinforcement Learning (RL) methods [Bello et al., 2017, Khalil et al., 2017, Kool et al., 2019, Ahn et al., 2020, Böther et al., 2022, Mazyavkina et al., 2021] for CO is an active field of research. In this work we are interested in training models that generate solutions at inference in contrast to approaches

---

[1] Our code is available at: `https://github.com/ml-jku/VAG-CO`.

that require problem instance specific training. A popular approach is to take a representation of the CO problem instance as input to a deep learning model and to output the parameters of a distribution over solutions that has its probability mass concentrated on regions with high solution quality. This probabilistic optimization approach to CO is used in the recent works of e.g. [Karalias and Loukas, 2020, Min et al., 2022, Qiu et al., 2022, Sun et al., 2022]. In these works the distribution over solutions is built on the assumption of mutual statistical independence of the individual solution parameters and can, consequently, be represented by a product of Bernoulli distributions for the the individual solution variables. This simplification is typically refereed to as a *mean-field approximation* (MFA) and is frequently used in various fields including e.g. statistical mechanics [Parisi, 1988], Bayesian statistics [Wainwright and Jordan, 2008], and the analysis of neural networks [Mei et al., 2019]. However, the simplifying assumption of the MFA restricts the expressivity of the corresponding distributions which limits its applicability when the target distribution represents strong correlations [Jaakkola and Jordan, 1998]. However, replacing MFA with more expressive approaches is computationally expensive and requires careful regularization to ensure efficient and stable training. Based on these consideration our contributions can be summarized as follows.

We demonstrate that the frequently used MFA imposes limits on the attainable solution quality for CO problems. Importantly, we introduce *Variational Annealing on Graphs for Combinatorial Optimization (VAG-CO)* a method that achieves new state-of-the-art performances on popular CO problems by combining expressive autoregressive models with annealed entropy regularization. We provide a theoretical motivation for this entropy regularization via considerations of the sample complexity of related density estimation problems. By introducing *sub-graph tokenization* VAG-CO drastically reduces the number of necessary steps to generate solutions and therein significantly improves the training and inference efficiency.

## 2    Problem Description

**Ising Formulation of CO.** To introduce VAG-CO it is convenient to adopt a point of view on CO that is motivated by statistical mechanics. As shown in Lucas [2014] many frequently encountered NP-complete CO problems can be reformulated in terms of a particular class of models known as Ising models which therefore allow the representation of different CO problem types in a unified manner. In this context the CO problem is equivalent to finding the minimum of an energy function $E : \{-1, 1\}^N \mapsto \mathbb{R}$. This function assigns an energy $E(\boldsymbol{\sigma})$ to an N-tuple of discrete solution variables $\boldsymbol{\sigma} = (\sigma_1 \ldots \sigma_N) \in \{-1, 1\}^N$ which are often referred to as spins. The family of aforementioned Ising models is characterized by the following form of the energy function:

$$E(\boldsymbol{\sigma}) = \sum_{i<j} J_{ij}\sigma_i\sigma_j + \sum_i^N B_i\sigma_i, \tag{1}$$

where the first term represents the interaction between spins $\sigma_i \in \{-1, 1\}$ through couplings $J_{ij} \in \mathbb{R}$, while the second term determines the magnitude $B_i \in \mathbb{R}$ of the contribution of an individual spin to the energy of the system. For brevity we will denote the parameters of the energy function $(J_{ij}, B_i)$ simply as $E$. Hence, $E$ will represent a CO problem instance while $E(\boldsymbol{\sigma})$ will denote the energy of a solution $\boldsymbol{\sigma}$ under an energy function with parameters $E$. The Ising formulation for the four CO problem types studied in this work is given in Tab. 1.

**Variational Learning.** Next, we specify how we approach the learning problem of approximating the optimal solution for a given problem instance. We follow the frequently taken variational learning approach of using a neural network with parameters $\theta$ to obtain for a given CO problem instance $E$ an associated probability distribution $p_\theta(\boldsymbol{\sigma}|E)$ over the space of possible solutions $\{-1, 1\}^N$. The problem instances $E$ are assumed to be independently sampled from a probability distribution $q(E)$ with $\Omega = \mathrm{supp}(q)$. In this setting the learning problem can be formulated as finding $\mathrm{argmin}_\theta \sum_\Omega q(E) \sum_{\boldsymbol{\sigma}} p_\theta(\boldsymbol{\sigma}|E)E(\boldsymbol{\sigma})$. Here $\sum_{\boldsymbol{\sigma}}$ is a shorthand for the sum over all $\boldsymbol{\sigma} \in \{-1, 1\}^N$. In practice both sums are approximated by empirical averages over samples from the corresponding distributions $q$ and $p_\theta$. In CO $E(\boldsymbol{\sigma})$ is typically characterized by many local minima. Therefore, directly approaching the learning problem described above via gradient descent methods is prone to getting stuck at sub-optimal parameters. As we demonstrate in Fig. 1 (Right) this problem can be alleviated by adding a regularization term to the optimization objective Eq. 1 which encourages $p_\theta$ to avoid a collapse to local minima. The resulting loss function $L : \{-1, 1\}^N \to \mathbb{R}$ associated to the

learning problem is given by:

$$L(\theta; \beta) = \frac{1}{M} \sum_{m=1}^{M} \hat{F}(\theta; \beta, E_m), \tag{2}$$

where $M$ is the number of CO problem instances $E_m$ in one batch and $\beta$ plays the role of a regularization coefficient which is frequently referred to as the inverse temperature $1/T$. The loss for an individual problem instance $E_m$ is based on the so-called free-energy $F(\theta; \beta, E_m)$:

$$F(\theta; \beta, E_m) = \sum_{\boldsymbol{\sigma}} p_\theta(\boldsymbol{\sigma}|E_m)\big(E_m(\boldsymbol{\sigma}) + 1/\beta \log p_\theta(\boldsymbol{\sigma}|E_m)\big). \tag{3}$$

In practice, we approximate the expectation value $F(\theta; \beta, E_m)$ with an empirical sample average $\hat{F}(\theta; \beta, E_m)$ that is based on $n_S$ samples $\boldsymbol{\sigma} \sim p_\theta(\boldsymbol{\sigma}|E_m)$. The term $E_m(\boldsymbol{\sigma})$ on the right-hand side of Eq. 3 encourages the concentration of $p_\theta(\boldsymbol{\sigma}|E_m)$ on low-energy solutions. Small values of the regularization parameter $\beta \in \mathbb{R}_{>0}$ encourage $p_\theta(\boldsymbol{\sigma}|E_m)$ to have a large entropy. For a given value of $\beta$ the free-energy in Eq. 3 is known to be minimized by the Boltzmann distribution associated to $E_m$ (see e.g. [MacKay, 2003]):

$$p_B(\boldsymbol{\sigma}|E_m, \beta) = \frac{\exp -\beta E_m(\boldsymbol{\sigma})}{\sum_{\boldsymbol{\sigma}'} \exp -\beta E_m(\boldsymbol{\sigma}')}. \tag{4}$$

Thus by minimizing $L$ the model learns to approximate $p_B(\boldsymbol{\sigma}|E_m, \beta)$ for a given $E_m$ and $\beta$. For $\beta \to \inf$ the $p_B(\boldsymbol{\sigma}|E_m, \beta)$ has its mass concentrated on the global minima of $E_m$ and for $\beta \to 0$ it approaches the uniform distribution [Mézard and Montanari, 2009]. It is, therefore, to be expected that the minimization of $L$ becomes harder at low temperatures, i.e. as $\beta \to \inf$, which opens the opportunity to a principled curriculum learning approach (Sec. 4). Based on these considerations we reformulate CO problems as the variational problem of approximating $p_B(\boldsymbol{\sigma}|E_m, \beta)$ in the limit $\beta \to \inf$ with a variational ansatz $p_\theta$ that has the variational parameters $\theta$. This problem can be formalized as $\text{argmin}_\theta \lim_{\beta \to \inf} L(\theta; \beta)$.

## 3  Variational Annealing on Graphs

Our method VAG-CO addresses CO as a variational learning problem on graphs. In particular, given a set of CO problem instances it learns to approximate the Boltzmann distribution of the corresponding Ising models (Sec. 2) with an autoregressive distribution. To obtain efficient training with this expressive model we apply an annealed entropy regularization. We formulate this learning problem in an RL setting and use PPO to train our model. To alleviate the lengthy sampling process of autoregressive approaches we introduce sub-graph tokenization which allows us to generate multiple solution variables at in a single step without loosing expressive power. We further improve the memory efficiency of our approach by dynamically pruning the graph which represents the CO problem instance.

**Autoregressive Solution Generation.** In the following we specify how we represent $p_\theta(\boldsymbol{\sigma}|E_i)$ and how to sample it, i.e. how to generate solutions $\boldsymbol{\sigma}$.

1. Draw a problem instance $E = (B_i, J_{ij}) \sim q(E)$ from the data distribution $q$ and construct a graph $G = (V, \mathcal{E})$ based on $E$. The nodes $\nu_i \in V$ correspond to the spins $\sigma_i$ and the set of edges $\mathcal{E}$ represents the non-zero components of $J_{ij}$. The graph $G$ is equipped with node features $x_i = [B_i, t_i]$ that are associated to its nodes $\nu_i$. Here $t_i$ is a four dimensional one-hot encoding which indicates the four possible states (I-IV) of a node $\nu_i$ namely whether the corresponding spin $\sigma_i$ is set to the value $+1$ (I) or $-1$ (II) or whether it is to be generated in the current step (III) or afterwards (IV). The edges between nodes $\nu_i$ and $\nu_j$ are associated with the scalar edge features $J_{ij}$.

2. Order the graph nodes according to the breadth-first search (BFS) algorithm. The $i$-th node in this ordering is denoted as $\sigma_i$. Now $i = 1$ and $t_i$ is set to the state (III) and all $t_{>i}$ are set to (IV).

3. A GNN is used to parameterize a Bernoulli distribution $p_\theta(\sigma_i|G) = \text{GNN}_\theta(G)$ from which the value $\pm 1$ of $\sigma_i$ is sampled. The state encoding variables of $G$ are updated by setting

$t_i$ associated to the graph is now updated accordingly and $t_{i+1}$ is set to (III). Now $i$ is incremented.

4. The previous step is repeated until the values of all $\sigma_i$ are set.

We note that at each step $i$ the graph $G$ depends on the problem instance $E$ and the already generated spins $\sigma_{<i}$. Therefore this procedure represents an autoregressive parametrization of a distribution over the space of possible solutions $\{-1, 1\}^N$. By denoting the graph $G$ at step $i$ as $G_i$ we get:

$$p_\theta(\boldsymbol{\sigma}|E) = \prod_{i=1}^{N} p_\theta(\sigma_i|\sigma_{<i}, E) = \prod_{i=1}^{N} p_\theta(\sigma_i|G_i). \tag{5}$$

This approach is more expressive than MFA and can therefore be expected to be better suited to approximate the typically complex Boltzmann distributions (Eq. 4) encountered in CO. Next we specify how we realize a stable training procedure by employing RL methods. The model architecture and hyperparameters are detailed in App. A.10.

**Reinforcement Learning Setting.** To explain how we train our model it is convenient to adopt an RL perspective where an episode corresponds to the solution generation procedure described above. For this we consider a Markov Decision Process (MDP) that is given by $(S, A, P, R)$. Here, $S$ is the set of possible states and the state at step $i$ is denoted by $s_i \in S$. At each step $s_i$ is represented by the graph $G_i$. Given a state $s_i$ an action $a_i$ that represents the assignments of the spin value $\sigma_i \in \{-1, 1\}$ is sampled from a probability distribution which is parameterised by the policy $p_\theta(\sigma_i|G_i)$. After sampling an action the reward $R_i(G_i, \beta)$ is computed according to an objective that is based on Eq. 3. We use the relation $F(\theta; \beta, E) = -\mathbb{E}_{\boldsymbol{\sigma} \sim p_\theta}[\sum_{i=1}^{N} R_i(G_i, \beta)]$ and define the reward at step $i$ as

$$R_i(G_i, \beta) = -\left[\Delta E_i + \frac{1}{\beta} \log p_\theta(\sigma_i|\sigma_{<i}, E)\right], \tag{6}$$

where $\Delta E_i = \sigma_i \left[\sum_{j<i} J_{ij}\sigma_j + B_i\right]$. With this definition maximising the reward is equivalent to minimizing the free-energy in Eq. 3. We approximate $F(\theta; \beta, E)$ with the empirical mean of $-\sum_{i=1}^{N} R_i(G_i, \beta)$ based on $n_S$ samples $\boldsymbol{\sigma} \sim p_\theta(\sigma_i|\sigma_{<i}, E)$. Finally, the state is changed deterministically, i.e. the spin $\sigma_i$ is set according to the sampled action $a_i$. The state update corresponds to the updated of $t_i$ and $t_{i+1}$ as described in step 3 of the solution generation procedure. Our model learns to solve this MDP via the popular PPO algorithm [Schulman et al., 2017] which is described in more detail in App. A.9.

**Annealing.** As discussed in Sec. 2 our training objective is based on Eq. 3 which contains an entropy regularization term. The strength of this regularization is determined by temperature $T$. At first, the temperature is kept constant at $T > 0$ for $N_{\text{warmup}}$ steps. Then, the temperature is slowly adapted for $N_{\text{anneal}}$ steps by following a predefined annealing schedule (see App. A.14) that converges to $T = 0$ as the end of training is reached. This reduction of the temperature throughout training is motivated from a theoretical point of view in Sec. 4.

**Subgraph Tokenization.** By introducing subgraph tokenization we decrease the number of forward passes per CO problem instance without sacrificing expressivity. Instead of modelling the probability for the two possible values $\pm 1$ of a single spin, we let the policy represent a probability distribution $p_\theta(\sigma_{i:i+k}|\sigma_{<i}, E)$ over all $2^k$ possible configurations of $k$ consecutive spins in the BFS ordering (step 2 of the solution generation procedure). We represent $p_\theta(\sigma_{i:i+k}|\sigma_{<i}, E)$ with a softmax function of a $2^k$ dimensional vector that is output by the GNN (App. A.10). Subgraph tokenization represents a modification of step 3 of the solution generation procedure and yields an improvement of the performance (Sec. 6).

**Dynamic Graph Pruning.** We note that once the spin value of $\sigma_i$ is sampled its interaction with adjacent spins $\sigma_j$ is fixed. Therefore, assuming that we have sampled the first spin $\sigma_1$, its interaction with an adjacent spin $\sigma_j$, that is yet to be generated, can be expressed as $B(1 \mapsto j)_j \sigma_j$, where we introduce $B(1 \mapsto j)_j = J_{1j}\sigma_1$. Therefore, we can immediately remove generated spins from the graph and update the node embeddings $x_j = [B_j, t_j]$ of adjacent nodes with $B_j \leftarrow B_j + B(i \mapsto j)_j$. Reducing the graph size during the solution generation process has the benefit of reducing the memory requirements when processing graphs with a GNN as in step 3 of the solution generation procedure.

# 4 Theoretical Motivation of Annealing

In the following we show that the concept of an annealed entropy regularization in Eq. 3 can be motivated based the sample complexity of density estimation for Boltzmann distributions.

Consider the problem of approximating the probability density function of a Boltzmann distribution $p_B(s; E, \beta)$, where $E : \mathbb{R}^N \to [0, 1]$ is a suitably normalized energy function. To solve this task we are given a set of samples $\mathcal{S} = \{s_1, \ldots, s_m\}$ which are independently drawn from the corresponding target Boltzmann distribution $p_B(s; E, \beta^*)$ at a fixed inverse temperature $\beta^*$. For brevity we will denote the distribution associated to $p_B(s; E, \beta)$ from now on as $p(\beta)$ where $E$ is suppressed since it is a fixed function. The empirical distribution corresponding to $\mathcal{S}$ will be denoted as $\hat{p}$. Further assume that we can evaluate $E(s_i)$ for all $s_i \in \mathcal{S}$.

In the present context a natural feasibility criterion for the estimated distribution is that the expectation value of the energy function for the estimated distribution should be compatible with the corresponding value $\mathbb{E}_{p(\beta*)}E(s)$ of the target distribution. We approach this problem with the maximum entropy principle [Jaynes, 1957]. Informally, this principle prescribes that among all feasible distributions one should prefer the one that yields the highest entropy. Since $\mathcal{S}$ contains only a finite number of samples we cannot determine $\mathbb{E}_{p(\beta*)}E(s)$ with arbitrary accuracy. As already stated in a similar form in [Dudík et al., 2007] one obtains with Hoeffding's inequality that with a probability of $1 - \delta$:

$$|\mathbb{E}_{p(\beta*)}E(s) - \mathbb{E}_{\hat{p}}E(s)| < \sqrt{\ln(2/\delta)/(2m)}. \tag{7}$$

As shown in [Kazama and Tsujii, 2003] the resulting maximum entropy optimization problem with the inequality constraint Eq. (7) is equivalent to the following regularized maximum likelihood problem over the family of Boltzmann distributions: $\min_\beta \mathcal{L}_{\hat{p}}(\beta) + \lambda\beta$, where $\mathcal{L}_{\hat{p}}(\beta) = -\mathbb{E}_{\hat{p}} \log p(\beta)$ is the cross-entropy between $\hat{p}$ and $p(\beta)$. Based on a closely related result in [Dudík et al., 2007] we obtain the following bound for the approximation of maximum entropy distributions:

**Remark 1.** *Assume a bounded energy function $E : \mathbb{R}^N \to [0, 1]$ and let $\hat{\beta}$ minimize $\mathcal{L}_{\hat{p}}(\beta) + \lambda|\beta|$ where $\lambda = \sqrt{\ln(2/\delta)/(2m)}$. Then with a probability at least $1 - \delta$:*

$$D_{KL}\big(p(\beta^*)||p(\hat{\beta})\big) \leqslant \frac{|\beta^*|}{\sqrt{m}}\sqrt{2\ln(2/\delta)}. \tag{8}$$

See App. A.15.2 for further details. According to Eq. 8 the sample complexity of approximating a maximum entropy distribution at an inverse temperature $\beta$ is in $O(\beta^2)$.

Curriculum learning is a machine learning paradigm in which the difficulty of training tasks is gradually increased as training proceeds [Bengio et al., 2009]. In view of Theorem 1 the entropy annealing in VAG-CO (Sec. 3) can be regarded as a principled curriculum learning approach if an increased sample complexity is regarded as an indication of a more difficult learning problem. In supervised learning tasks Wu et al. [2021] find that the application of curriculum learning results in more resource efficient training but not necessarily in better performance. As we demonstrate in Sec. 6 our entropy annealing does actually yield better performing models.

# 5 Related Work

As pointed out in [Yehuda et al., 2020] supervised training of CO solvers faces the problem of expensive data generation and that data augmentation cannot circumvent this problem. Consequently, there is a growing interest in RL and unsupervised methods. In the following we focus on methods that attempt to learn how to generate solutions rather than how to improve existing ones.

**Unsupervised Learning and Reinforcement Learning.** The work of [Bello et al., 2017] proposes to use an actor-critic approach to learn how to solve CO problems. They were the first to show that deep RL is a promising approach for CO. Their method represents an autoregressive distribution over solutions. However, in their setting rewards are sparse since they are only available for complete solutions. In our approach rewards are dense since they are available after every state transition. Another influential work is [Khalil et al., 2017] who were the first to exploit the graph structure of CO problems by learning problem representations with GNNs. Applying GNNs to CO problem has become a common choice since then [Cappart et al., 2021]. In [Tönshoff et al., 2020] the method

RUN-CSP is introduced. The work of [Karalias and Loukas, 2020] proposes Erdős Goes Neural (EGN) in which the goal is to minimize a loss that can be regarded as a certificate of the existence of a CO problem solution whose cost is upper bounded by the loss. By relying on an MFA they can calculate this loss efficiently. Also [Qiu et al., 2022] use an MFA to optimize a distribution over the solution space and optimize the corresponding parameters with gradient estimates that are obtained by REINFORCE. The work of [Min et al., 2022] uses the same MFA-based concept as EGN and focuses on alleviating the oversmoothing problem of GNNs by using modified Graph Convolutional Networks [Kipf and Welling, 2017]. An approach based on an entry-wise concave continuous relaxation of the discrete CO loss and a subsequent rounding procedure for integral solution generation is introduced in [Wang et al., 2022]. Here the concept of [Karalias and Loukas, 2020] is generalized to a wider class of problems and rather simple rounding procedure is introduced to generate solutions efficiently. This approach is further extended by [Wang and Li, 2023] to a meta-learning setting called Meta-EGN in which network parameters are updated for individual CO problem instances. They also argue that RL based CO methods suffer from unstable training. We call this claim into question by finding no stability issues with our RL-based method. Further VAG-CO outperforms EGN and even Meta-EGN despite not updating any parameters on test problem instances. The approach of extending functions on sets to continuous supports is taken in Neural Set Function Extensions (NSFE) [Karalias et al., 2022]. In NSFE the discrete CO loss function is replaced with a convex combination of the discrete loss function values at certain integral solutions. These extensions can be regarded as expectation values that can be calculated in closed form without sampling. Whether the lack of sampling noise in NSFE is beneficial for the optimization procedure is not obvious.

**Variational Annealing.** The concept of using autoregressive models in the variational problem of approximating Boltzmann distributions of Ising models was introduced by [Wu et al., 2019]. They show that variational annealing (VA), i.e. the combination of the variational approach and temperature annealing, is a highly performant method. The work of [Hibat-Allah et al., 2021] compares VA to other ones like Simulated Annealing (SA) [Kirkpatrick et al., 1983] and confirms its strong performance on problems related to spin glasses. In [Khandoker et al., 2023] the strong performance of VA compared to SA is confirmed on CO problems. A crucial aspect of these works on VA with respect to ours and the ones in the previous paragraph is that they optimize the parameters of their models only for individual problem instances. They do not attempt to learn how to generalize over a family of problems. The work of [Sun et al., 2022] aims at a generalizing application of VA in CO by using an MFA. Our experiments indicate that the simplicity of MFA-based methods leads to performance limitations in particular on hard CO problem instances.

# 6 Experiments

We evaluate VAG-CO on various CO problems that are studied in the recent literature. Additionally, we evaluate VAG-CO on synthetic datasets where solving the corresponding CO problem is known to be particularly hard. Finally, we discuss experiments on the impact of entropy regularization and subgraph tokenization. Our result tables also include inference runtimes. A quantitative runtime comparison is, however, difficult since the runtimes reported from other works and for Gurobi 10.0.0 [Gurobi Optimization, LLC, 2023] were obtained with differen setups. See App. A.12 for details on the runtime measurements. For Gurobi we report results for various different runtime limits.

**Maximum Independent Set.** In the following we will compare VAG-CO on the Maximum Independent Set (MIS) problem to recently published results from [Karalias et al., 2022], where the graph datasets COLLAB, ENZYMES and PROTEINS Morris et al. [2020] are used. In the MIS problem the task is to find the largest subset of independent, i.e. unconnected, nodes in a graph. As an optimization objective for VAG-CO we use Eq. 2 with the Ising energy function for MIS (see Tab. 1). Here, the energy function $E(\boldsymbol{q})$ consists of two terms $E_A(\boldsymbol{q})$ and $E_B(\boldsymbol{q})$, that depend on the binary representation of the solution $\boldsymbol{q} = \frac{\sigma+1}{2}$ with $\boldsymbol{q} \in \{0,1\}^N$. When $q_i = 1$ the corresponding node is defined to be included in the set and it is excluded otherwise. The first term $E_A(\boldsymbol{q})$ is proportional to the number of nodes that are in the set, whereas $E_B(\boldsymbol{q})$ is proportional to the number of independence violations, i.e. the number of connected nodes within the set. By selecting $A, B \in \mathbb{R}_+$ such that $A < B$ we ensure that all minima satisfy $E_B = 0$ since in this case excluding a violating node from the set always reduces the energy. In our experiments, we choose $A = 1.0$ and $B = 1.1$.

We follow the procedure of [Karalias et al., 2022] and use a $0.6/0.1/0.3$ train/val/test split on the aforementioned datasets and use only the first 1000 graphs on the COLLAB dataset. The results

| Problem Type | Ising formulation: $\min_q E(\boldsymbol{q})$ where $E(\boldsymbol{q}) = E_A(\boldsymbol{q}) + E_B(\boldsymbol{q})$ |
|---|---|
| MVC | $E(\boldsymbol{q}) = A \sum_i^N q_i + B, \sum_{(i,j)\in\mathcal{E}}(1-q_i)\cdot(1-q_j)$ |
| MIS | $E(\boldsymbol{q}) = -A \sum_i^N q_i + B \sum_{(i,j)\in\mathcal{E}} q_i \cdot q_j$ |
| MaxCl | $E(\boldsymbol{q}) = -A \sum_i^N q_i + B \sum_{(i,j)\notin\mathcal{E}} q_i \cdot q_j$ |
| MaxCut | $E(\boldsymbol{\sigma}) = -\sum_{(i,j)\in\mathcal{E}} \frac{1-\sigma_i\sigma_j}{2}$    where $\sigma_i = 2\,q_i - 1$ |

Table 1: Ising formulations of the MVC, MIS, MaxCl and MaxCut problem ([Lucas, 2014]). The term that includes the constant $A$ corresponds to $E_A(q)$ and the term that includes the constant $B$ corresponds to $E_B(q)$.

|  | ENZYMES | PROTEINS | IMDB-BINARY | COLLAB | MUTAG |
|---|---|---|---|---|---|
| Evaluation metric A.1 | $AR^*$ (s/graph ) | $AR^*$ (s/graph ) | $AR^*$ (s/graph ) | $AR^*$ (s/graph ) | $AR^*$ (s/graph ) |
| EGN (r) | $0.821 \pm 0.124$ (N/A) | $0.903 \pm 0.114$ (N/A) | $0.515 \pm 0.310$ (N/A) | $0.886 \pm 0.198$ (N/A) | $0.939 \pm 0.069$ (N/A) |
| NSFE (r) | $0.775 \pm 0.155$ (N/A) | $0.729 \pm 0.205$ (N/A) | $0.679 \pm 0.287$ (N/A) | $0.392 \pm 0.253$ (N/A) | $0.854 \pm 0.132$ (N/A) |
| REINFORCE (r) | $0.751 \pm 0.301$ (N/A) | $0.725 \pm 0.285$ (N/A) | $0.881 \pm 0.240$ (N/A) | $1.000$ (N/A) | $0.781 \pm 0.316$ (N/A) |
| Straight-through (r) | $0.725 \pm 0.268$ (N/A) | $0.722 \pm 0.26$ (N/A) | $0.917 \pm 0.253$ (N/A) | $0.856 \pm 0.221$ (N/A) | $0.965 \pm 0.162$ (N/A) |
| DB-Greedy | $0.9810 \pm 0.0009$ (0.008) | $0.9848 \pm 0.0003$ (0.008) | $1.000$ (0.007) | $0.9968 \pm 0.0001$ (0.033) | $0.994 \pm 0.001$ (0.005) |
| MFA: CE | $0.9875 \pm 0.001$ (0.036) | $0.9883 \pm 0.0007$ (0.037) | $1.000$ (0.022) | $0.9985 \pm 0.0004$ (0.054) | $1.000$ (0.019) |
| MFA-Anneal: CE | $0.9880 \pm 0.0014$ (0.036) | $0.9881 \pm 0.0007$ (0.037) | $1.000$ (0.022) | $\mathbf{0.9991 \pm 0.0004}$ (0.054) | $1.000$ (0.019) |
| VAG-CO (ours) | $\mathbf{0.9960 \pm 0.0007}$ (0.026) | $\mathbf{0.9977 \pm 0.0005}$ (0.032) | $1.000$ (0.017) | $0.9988 \pm 0.0002$ (0.064) | $1.000$ (0.015) |
| Gurobi ($t_{\max} = 0.01$) | $1.000$ (0.001) | $0.9996 \pm 0.0001$ (0.001) | $1.000$ (0.0003) | $0.997 \pm 0.001$ (0.002) | $1.000$ (0.0001) |
| Gurobi ($t_{\max} = 0.1$) | $1.000$ (0.001) | $1.000$ (0.001) | $1.000$ (0.0003) | $1.000$ (0.002) | $1.000$ (0.0001) |
| Evaluation metric A.1 | $\widehat{AR}$ (s/graph ) | $\widehat{AR}$ (s/graph ) | $\widehat{AR}$ (s/graph ) | $\widehat{AR}$ (s/graph ) | $\widehat{AR}$ (s/graph ) |
| MFA | $0.9635 \pm 0.0014$ (0.0029) | $0.9705 \pm 0.0012$ (0.0025) | $0.990 \pm 0.002$ (0.0028) | $0.960 \pm 0.007$ (0.0024) | $0.970 \pm 0.003$ (0.0028) |
| MFA-Anneal | $0.968 \pm 0.001$ (0.0029) | $0.973 \pm 0.002$ (0.0025) | $0.994 \pm 0.001$ (0.0028) | $0.917 \pm 0.045$ (0.0024) | $0.975 \pm 0.002$ (0.0028) |
| VAG-CO (ours) | $\mathbf{0.9863 \pm 0.0007}$ (0.026) | $\mathbf{0.9908 \pm 0.0005}$ (0.032) | $\mathbf{0.995 \pm 0.001}$ (0.017) | $\mathbf{0.993 \pm 0.001}$ (0.064) | $\mathbf{0.994 \pm 0.002}$ (0.015) |

Table 2: Results on the MIS Problem (see Sec. 1). We show test set results on the best approximation ratio $AR^*$ and the average approximation ratio $\widehat{AR}$ across different methods and datasets. Values that are closer to one are better. (r) indicates that these are results as reported in [Karalias et al., 2022]. The time for each algorithm is reported in round brackets as seconds per graph.

are shown in Tab. 2 where the test set average of the best approximation ratio $AR^*$ out of $n_S = 8$ sampled solutions per graph is reported (see App. A.1). This metric was originally proposed in [Karalias and Loukas, 2020]. We also report results of our own implementation of an MFA-based method that is trained with REINFORCE. This method is used with (MFA-Anneal) and without (MFA) annealing (App. A.5). As in Wang and Li [2023] and Karalias and Loukas [2020] we use the conditional expectation procedure (CE, App. A.5.5) to sample solutions and report the corresponding results as (MFA: CE) and (MFA-Anneal: CE). We also add results obtained by the Degree Based Greedy (DB-Greedy) algorithm [Wormald, 1995] that is proposed by [Angelini and Ricci-Tersenghi, 2023] as a baseline for machine learning algorithms on MIS. For VAG-CO we greedily sample different states for a given problem instance by using different BSF orderings of the corresponding graph nodes. Our results show that VAG-CO significantly outperforms all competitors including the MFA on ENZYMES and PROTEINS. On IMDB-BINARY and MUTAG the MFA-based approaches and VAG-CO outperform all other machine learning methods and achieve an optimal $AR^*$. On COLLAB MFA-Anneal and VAG-CO exhibit the best results with insignificantly better results for MFA-Anneal. We also report results based on the test set average of the approximation ratio with $n_S = 30 \ \widehat{AR}$ (App. A.1). This metric shows the performance of the learned probability distribution for each model, when no post processing is applied. Our results show that VAG-CO always achieves a significantly better performance in terms of $\widehat{AR}$ than MFA-based approaches.

**Minimum Vertex Cover.** We compare VAG-CO to results from Wang and Li [2023] where the Minimum Vertex Cover (MVC) problem is solved on the TWITTER [Leskovec and Krevl, 2014], COLLAB and IMDB-BINARY Morris et al. [2020] graph datasets. The MVC problem is the task of finding the smallest subset of vertices such that each edge has at least one node in the subset. As for the MIS problem we formulate this CO problem in terms using the Ising energy function that is defined in Tab. 1 and set $A = 1.0$ and $B = 1.1$. We follow the procedure of Wang and Li [2023] and use a $0.7/0.1/0.2$ train/val/test split on these datasets. Their method Meta-EGN uses meta

learning in combination with the EGN method. They apply CE to generate solutions and report the best out of $n_S = 8$ sampled solutions. Additionally, they report fine tuning results denoted by (f-t), where they apply one step of fine tuning individually for each CO problem instance in the test dataset. Our results in Tab. 3 show that VAG-CO improves upon the results reported in Wang and Li [2023] across all datasets. We also show that the MFA with and without annealing performs remarkably well on the TWITTER and COLLAB dataset and even outperforms VAG-CO on the COLLAB dataset. The results in terms of $\widehat{\text{AR}}$ show that the strong performance of the MFA can be attributed to the post-processing method CE rather than to the learned MFA-based model.

**Maximum Clique.** The Maximum Clique (MaxCl) Problem is the problem of finding the largest set of nodes, in which each node is connected to all other nodes that are contained in the set. The MaxCl problem is equivalent to solving the MIS problem on the complementary graph. The complementary graph can be obtained by connecting nodes that are not connected in the original graph and by disconnecting nodes that are connected. We evaluate our method on the ENZYMES and IMDB-BINARY datasets with the same train/val/test splits

|                              | TWITTER | COLLAB | IMDB-BINARY |
|------------------------------|---------|--------|-------------|
| Evaluation metric A.1        | $AR$* (s/graph) | $AR$* (s/graph) | $AR$* (s/graph) |
| EGN (r)                      | $1.033 \pm 0.023$ (0.29) | $1.013 \pm 0.022$ (0.15) | $1.000$ (0.08) |
| EGN f-t (r)                  | $1.028 \pm 0.021$ (0.80) | $1.008 \pm 0.015$ (0.38) | $1.000$ (0.32) |
| RUN-CSP (r)                  | $1.180 \pm 0.435$ (0.16) | $1.208 \pm 0.198$ (0.19) | $1.188 \pm 0.178$ (0.08) |
| Meta-EGN (r)                 | $1.019 \pm 0.017$ (0.29) | $1.003 \pm 0.010$ (0.15) | $1.000$ (0.08) |
| Meta-EGN f-t (r)             | $1.017 \pm 0.017$ (0.80) | $1.002 \pm 0.010$ (0.38) | $1.000$ (0.32) |
| Greedy (r)                   | $1.014 \pm 0.014$ (1.95) | $1.209 \pm 0.198$ (1.79) | $1.180 \pm 0.077$ (0.02) |
| MFA: CE                      | $1.0064 \pm 0.0004$ (0.08) | $\mathbf{1.00021 \pm 0.00003}$ (0.05) | $1.000$ (0.02) |
| MFA-Anneal: CE               | $1.0041 \pm 0.0001$ (0.08) | $1.00023 \pm 0,00004$ (0.05) | $1.000$ (0.02) |
| VAG-CO (ours)                | $\mathbf{1.0024 \pm 0.00006}$ (0.15) | $1.0017 \pm 0.0002$ (0.16) | $1.000$ (0.016) |
| Gurobi ($t_{\max} = 0.01$)   | $1.054 \pm 0.001$ (0.008) | $1.0002 \pm 0.0001$ (0.002) | $1.000$ (0.0003) |
| Gurobi ($t_{\max} = 0.1$)    | $1.0015 \pm 0.00001$ (0.025) | $1.000$ (0.004) | $1.000$ (0.0003) |
| Gurobi ($t_{\max} = 0.2$)    | $1.0001 \pm 0.0001$ (0.028) | $1.000$ (0.004) | $1.000$ (0.0003) |
| Evaluation metric A.1        | $\widehat{AR}$ (s/graph) | $\widehat{AR}$ (s/graph) | $\widehat{AR}$ (s/graph) |
| MFA:                         | $1.0106 \pm 0.0002$ (0.004) | $1.0106 \pm 0.0001$ (0.003) | $1.0014 \pm 0.0004$ (0.003) |
| MFA-Anneal:                  | $1.0090 \pm 0.0004$ (0.004) | $\mathbf{1.0048 \pm 0.0002}$ (0.003) | $1.0011 \pm 0.0002$ (0.003) |
| VAG-CO (ours)                | $\mathbf{1.0079 \pm 0.0003}$ (0.15) | $1.0056 \pm 0.0002$ (0.16) | $1.0011 \pm 0.0006$ (0.016) |

Table 3: Results on the MVC problem (see Sec. 1). We show test set results on the best approximation ratio $AR$* and the average approximation ratio $\widehat{AR}$ across different methods and datasets. Values that are closer to one are better. (r) indicates that these results are reported in [Wang and Li, 2023].

that are used in the MIS problem (see Sec. 6). Results are shown in Tab. 4, where we compare to results that are reported in [Karalias et al., 2022] and to our own implementation of MFA and MFA-Anneal. Here, we see that VAG-CO is always among the best performing method in therms of $AR$* and always the single best method in therms of $\widehat{AR}$.

**Maximum Cut.** The Maximum Cut (MaxCut) problem is the problem of finding two sets of nodes so that the number of edges between these two sets is as high as possible. We evaluate our method on BA graphs [Barabási and Albert, 1999], where we follow [Zhang et al., 2023] and generate 4000/500/500 train/val/test graphs between the size of 200-300 nodes and set the generation parameter $m$ to 4. Results on this dataset are shown in Tab. 4, where we compare our method to MaxCut Values ($MCut = \sum_{(i,j) \in \mathcal{E}} \frac{1 - \sigma_i \sigma_j}{2}$) that are reported in [Zhang et al., 2023]. Our method achieves the best $\widehat{MCut}$ results among all learned methods.

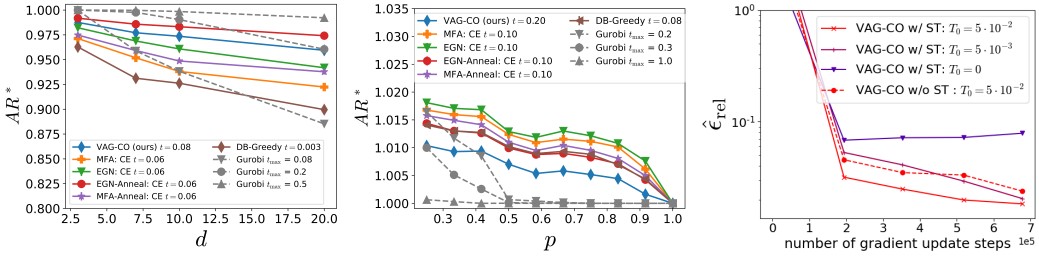

Figure 1: Left: Comparison of $AR$* for the MIS problem on the RRG-100 dataset with different degrees $d$. Middle: Results for the MVC problem in terms of $AR$* on the RB-200 dataset at different generation parameters $p$. Left, Middle: Values closer to one are better. The runtime $t$ is given in s/graph and $t_{\max}$ is the time limit for Gurobi. Right: Ablation on annealing and subgraph tokenization (ST). The $\hat{\epsilon}_{\text{rel}}$ on the RB-100 MVC is plotted over the number of gradient update steps. Lower values are better.

| Evaluation metric A.1 | ENZYMES $AR^*$ (s/graph) | IMDB-BINARY $AR^*$ (s/graph) |
|---|---|---|
| EGN (r) | $0.883 \pm 0.156$ (N/A) | $0.936 \pm 0.175$ (N/A) |
| NSFE (r) | $0.933 \pm 0.148$ (N/A) | $0.961 \pm 0.143$ (N/A) |
| REINFORCE (r) | $0.751 \pm 0.301$ (N/A) | $0.881 \pm 0.240$ (N/A) |
| Straight-through (r) | $0.725 \pm 0.268$ (N/A) | $0.917 \pm 0.253$ (N/A) |
| DB-Greedy | $0.9485 \pm 0.0034$ (0.025) | $0.9875 \pm 0.0008$ (0.01) |
| MFA: CE | $0.9766 \pm 0.001$ (0.04) | $\mathbf{0.999 \pm 0.009}$ (0.025) |
| MFA-Anneal: CE | $\mathbf{0.9921 \pm 0.0017}$ (0.04) | $\mathbf{0.999 \pm 0.0001}$ (0.025) |
| VAG-CO (ours) | $\mathbf{0.987 \pm 0.004}$ (0.029) | $\mathbf{0.9981 \pm 0.0013}$ (0.017) |
| Gurobi ($t_{\max} = 0.01$) | $0.983 \pm 0.002$ (0.004) | $0.996 \pm 0.0002$ (0.001) |
| Gurobi ($t_{\max} = 0.1$) | 1.000 (0.005) | 1.000 (0.002) |
| Evaluation metric A.1 | $\widehat{\text{AR}}$ (time in [s/graph]) | $\widehat{\text{AR}}$ (time in [s/graph]) |
| MFA | $0.764 \pm 0.013$ (0.001) | $0.983 \pm 0.009$ (0.001) |
| MFA-Anneal | $0.806 \pm 0.004$ (0.001) | $0.974 \pm 0.004$ (0.001) |
| VAG-CO (ours) | $\mathbf{0.943 \pm 0.012}$ (0.029) | $\mathbf{0.993 \pm 0.0013}$ (0.017) |

| | BA $200 - 300$ $\widehat{\text{MCut}}$ (s/graph) |
|---|---|
| Gurobi (r) | 732.47 (1.57) |
| SDP (r) | 700.36 (4.29) |
| Greedy (r) | 688.31 (0.026) |
| EGN (r) | 693.45 (0.092) |
| Anneal (r) | 696.73 (0.09) |
| GFlowNet (r) | 704.30 (0.35) |
| VAG-CO (ours) | $\mathbf{722.31}$ (0.17) |
| Gurobi ($t_{\max} = 0.1$) | 731.34 (0.1) |
| Gurobi ($t_{\max} = 1.$) | 731.79 (1.) |
| Gurobi ($t_{\max} = 2.$) | 732.00 (2.) |
| Gurobi ($t_{\max} = 10.$) | 732.01 (10.) |

Table 4: Left: Results on the Maximum Clique Problem (see Tab. 1). We show test set results on the best approximation ratio $AR^*$ and the average approximation ratio $\widehat{\text{AR}}$ across different methods and datasets. Values that are closer to one are better. (r) indicates that these are results as reported in [Karalias et al., 2022]. The time for each algorithm is reported in round brackets as seconds per graph. Right: Results on the Maximum Cut Problem (see Tab. 1) on BA $200 - 300$ graphs. We show test set results on the average maximum cut value $\widehat{\text{MCut}}$ across different methods. Values that are larger are better. (r) indicates that these are results as reported in [Zhang et al., 2023].

**Evaluation on Synthetic Problems for MIS.** On many of the benchmarks above nearly optimal results are obtained. Therefore, we conduct additional experiments on synthetic graph datasets that yield hard CO problems.

For graphs with a degree $d$ larger than 16 the MIS problem on random regular graphs (RRGs) is known to be particularly hard [Barbier et al., 2013]. Therefore, we generate RRGs with an average size of 100 nodes (RRG-100), where we sample 4100 RRGs between the size of 80 to 120 with different degrees $d \in [3, 7, 10, 20]$. For training, validation and testing we use $3000/100/1000$ graphs. Results for the MIS problem on the RRG-100 dataset are shown in Fig. 1 (Left), where we plot the test set average of the best relative error $\epsilon^*_{\text{rel}}$ out of $n_S = 8$ sampled solutions per problem instance. Here the relative error is calculated with respect to exact solutions that are obtained with the commercial solver Gurobi. Error bars indicate the standard error over graphs with the corresponding node degree. As expected we find that for all methods $\epsilon^*_{\text{rel}}$ increases for larger degrees. MFA and MFA-Anneal outperform the DB-Greedy method. Furthermore, VAG-CO is the best performing method for all graph degrees. We also show Gurobi performance for each $d$ at different time limits $t_{\max}$. On this dataset VAG-CO outperforms Gurobi with the given compute budged (see App. A.12) by a high margin on hard instances.

**Evaluation on Synthetic Problems for MVC.** We also conduct experiments on graphs that are generated by the so-called RB method [Xu et al., 2005]. This method allows the generation of graphs that are known to yield hard MVC problem instances [Wang and Li, 2023]. The RB model has three distinct generation parameters: $n, k'$, and $p$. Adjusting the values of $n$ and $k'$ allows us to control the expected size of the generated graphs, while $p$ serves as a parameter to regulate the hardness of the graph instances. Specifically, when $p$ is close to one, the generated graphs tend to be easier, whereas reducing $p$ leads to increased difficulty. To ensure diversity in the RB dataset, we generate graphs with varying levels of hardness by randomly sampling $p$ from the range of 0.25 to 1.0. For our experiments, we utilize the RB-200 dataset, consisting of RB graphs with an average size of 200 nodes with a train/val/test split of $2000/500/1000$ graphs. Following [Wang and Li, 2023] each graph in this dataset is generated by randomly selecting values for $n$ from the range of 20 to 25, and $k'$ from the range of 9 to 10. Figure 1 (Middle) shows the corresponding results in terms of $AR^*$ in dependence

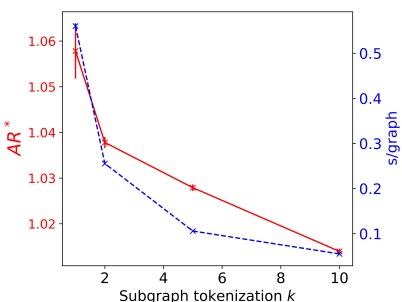

Figure 2: Study on the advantages of larger ST configuration sizes $k$ for VAG-CO on the RB-100 MVC dataset. On the left y-axis we plot $AR^*$ (lower values are better) for VAG-CO models that are trained for values of $k$. The left y-axis shows the inference time per graph.

of $p$. We find that the DB-Greedy algorithm outperforms both MFA approaches. The MFA with annealing tends to perform worse than the MFA without annealing on this dataset. Here, we also add our own implementation of EGN and EGN-Anneal [Sun et al., 2022] and find that EGN performs similarly to MFA and EGN-Anneal achieves a similar performance as DB-Greedy. VAG-CO outperforms all other methods for all values of $p$ on this dataset. We also show Gurobi performance for each $p$ at different time limits $t_{\max}$. Here, we see that at high $p$ values Gurobi achieves close to optimal results within a very small time budged and is the best performing method. Whereas for low p values Gurobi performance drops by a high margin and VAG-CO achieves similar results within a comparable time.

**Ablation on Annealing.** In the following we investigate whether the annealed entropy regularization in Eq. 3 is indeed beneficial. These experiment are conducted for the MVC problem on RB-100 graphs that are generated with $n \in [9, 15]$, $k^{'} \in [8, 11]$ and $p \in [0.25, 1]$. We compare VAG-CO test set learning curves with three different initial temperatures $T_0 \in \{5 \cdot 10^{-2}, 5 \cdot 10^{-3}, 0\}$. The run with $T_0 = 0$ is equivalent to VAG-CO without annealing (App. A.14). Figure 1 (right) shows the average relative error $\hat{\epsilon}_{\mathrm{rel}}$ for $n_S = 30$ sampled solutions per problem instance on the test set over the number of gradient update steps. We find that the run without annealing starts to converge rather quickly but at a comparably bad relative energy. In contrast to that, the runs with annealing ($T_0 \in \{5 \cdot 10^{-2}, 5 \cdot 10^{-3}\}$) keep improving over more steps and achieve a better final performance.

**Ablation on Subgraph Tokenization.** Next we study the impact of subgraph tokenization (see Sec. 3) on the performance of VAG-CO. The same problem setting as in the annealing ablation above is used. Figure 1 (right) compares VAG-CO with the default $k = 5$ subgraph tokenization (w/ ST) to VAG-CO without subgraph tokenization (w/o ST). For a fair comparison the numbers of gradient update steps and the annealing schedules are set to be equal. We find that with subgraph tokenization we achieve a considerably better $\hat{\epsilon}_{\mathrm{rel}}$. These results underpin that subgraph tokenization does indeed yield an improved efficiency in terms of gradient update steps.

We additionally provide a more detailed investigation on ST, where we compare VAG-CO with four different values of $k \in \{1, 2, 5, 10\}$ on the RB-100 MVC dataset. In our experiments we keep the number of gradient update steps constant and iteratively tune for the best learning rate and the best initial temperature in each setting of $k$. Results are shown in Fig. 2, where $AR^*$ and the average time per graph are plotted over $k$. As $k$ increases the performance in terms of $AR^*$ does improve and the inference time is reduced considerably. This experiment underscores the performance and scalability improvement due to ST.

# 7 Limitations

While MFA-based approaches typically generate a solution to a CO problem in a single forward pass VAG-CO requires a number of forward passes that is proportional the problem size $N$. However, methods with solution generation in a single forward pass frequently rely on post-processing procedures like CE that have a time complexity which is also linear in $N$. In our approach the nodes in the graph are always processed in an order that is determined by a BFS. Therefore studying the impact of other node orderings or making the algorithm invariant to such orderings might be an interesting future research direction. For the annealing schedule we report comparisons for VAG-CO with three different initial temperatures but a comprehensive study on the optimization of the annealing schedule is left to future investigations.

# 8 Conclusion

Our results show that learning to solve hard CO problems requires sufficiently expressive approaches like autoregressive models. With VAG-CO we introduce a method that enables stable and efficient training of such a model through an annealed entropy regularization. We provide a theoretical motivation for this regularization method and demonstrate its practical benefit in experiments. In addition, we demonstrate that by systematically grouping solution variables VAG-CO achieves an improved performance with respect to a naive autoregressive model. Importantly VAG-CO outperforms recent approaches on numerous benchmarks and exhibits superior performance on synthetic CO problems that are designed to be hard.

# 9 Acknowledgements

We thank Günter Klambauer for useful discussions and comments . The ELLIS Unit Linz, the LIT AI Lab, the Institute for Machine Learning, are supported by the Federal State Upper Austria. We thank the projects AI-MOTION (LIT-2018-6-YOU-212), DeepFlood (LIT-2019-8-YOU-213), Medical Cognitive Computing Center (MC3), INCONTROL-RL (FFG-881064), PRIMAL (FFG-873979), S3AI (FFG-872172), DL for GranularFlow (FFG-871302), EPILEPSIA (FFG-892171), AIRI FG 9-N (FWF-36284, FWF-36235), AI4GreenHeatingGrids(FFG- 899943), INTEGRATE (FFG-892418), ELISE (H2020-ICT-2019-3 ID: 951847), Stars4Waters (HORIZON-CL6-2021-CLIMATE-01-01). We thank Audi.JKU Deep Learning Center, TGW LOGISTICS GROUP GMBH, Silicon Austria Labs (SAL), FILL Gesellschaft mbH, Anyline GmbH, Google, ZF Friedrichshafen AG, Robert Bosch GmbH, UCB Biopharma SRL, Merck Healthcare KGaA, Verbund AG, GLS (Univ. Waterloo) Software Competence Center Hagenberg GmbH, TÜV Austria, Frauscher Sensonic, TRUMPF and the NVIDIA Corporation.

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

# A  Appendix

## A.1  Evaluation Metrics

We use different metrics to evaluate the performance of models on CO problems. We assess the quality of an individual solution $\boldsymbol{\sigma}$ by the associated value of the energy function $E_m(\boldsymbol{\sigma})$ which represents the size of the solution sets in MIS and MVC. Here $m$ refers to the problem instance under consideration. Optimal solutions $\boldsymbol{\sigma}_{\mathrm{opt}}$ are obtained with the Gurobi solver [Gurobi Optimization, LLC, 2023]. For models that generate solutions indeterministically we sample $n_S$ different solutions $\boldsymbol{\sigma}_j$ per problem instance and calculate the test dataset average of the best relative error $\epsilon_{\mathrm{rel}}^*$:

$$\epsilon_{\mathrm{rel}}^* = \frac{1}{M} \sum_{m=1}^{M} \min_j \frac{|E_m(\boldsymbol{\sigma}_{opt}) - E_m(\boldsymbol{\sigma}_j)|}{|E_m(\boldsymbol{\sigma}_{\mathrm{opt}})|}, \tag{9}$$

where $M$ denotes the number of problem instances in the test dataset. In case of deterministic algorithms only one sample is generated, i.e. $n_S = 1$. In several experiments it is insightful to investigate the average relative error $\hat{\epsilon}_{\mathrm{rel}}$ for which we take the average instead of the minimum in Eq. 9. Analogously, we also define the best approximation ratio AR* by

$$\mathrm{AR}^* = \frac{1}{M} \sum_{m=1}^{M} \min_j \frac{|E_m(\boldsymbol{\sigma}_j)|}{|E_m(\boldsymbol{\sigma}_{\mathrm{opt}})|}, \tag{10}$$

The average approximation ratio $\widehat{\mathrm{AR}}$ is defined by taking the average instead of the minimum operation in Eq. 10. For VAG-CO we calculate the average over the ordered greedy (OG) sampling method (See. App A.7). We always report the evaluation metric together with the standard error over three different seeds except for the experiment on RRG-100 MIS, where only results on one seed are reported.

## A.2  Linear Integer Program Formulations

For Gurobi we formulate the CO Problems studied in this paper in Linear Integer Program Formulation (LIP) (see Tab. 5).

| Problem Type | ILP formulation $q \in \{0,1\}^N$ |
|:---:|:---|
| MVC | $\min\limits_{\boldsymbol{q}} \ \sum\limits_{i=1}^{N} q_i \quad \text{s.t.} \quad q_i + q_j \geqslant 1 \text{ if } (i,j) \in \mathcal{E}$ |
| MIS | $\min\limits_{\boldsymbol{q}} \ -\sum\limits_{i=1}^{N} q_i \quad \text{s.t.} \quad q_i + q_j \leqslant 1 \text{ if } (i,j) \in \mathcal{E}$ |
| MaxCl | $\min\limits_{\boldsymbol{q}} \ \sum\limits_{i=1}^{N} q_i \quad \text{s.t.} \quad q_i + q_j \leqslant 1 \text{ if } (i,j) \notin \mathcal{E}$ |
| MaxCut | $\max\limits_{e \in \{0,1\}^{|\mathcal{E}|}, q} \ \sum\limits_{(i,j) \in \mathcal{E}}^{N} e_{ij}$ 
 $\text{s.t.} \quad e_{ij} \leqslant q_i + q_j \text{ if } (i,j) \in \mathcal{E}$ 
 $\qquad\quad e_{ij} \leqslant 2 - (q_i + q_j) \text{ if } (i,j) \in \mathcal{E}$ |

Table 5: Integer Linear Program (ILP) formulations of MIS, MVC, MaxCl and MaxCut.

## A.3  Ensuring Feasible Solutions

Since there is no rigorous guarantee that the model samples only feasible solutions that satisfy the constraints, we use a fast post processing procedure to make sure that only feasible solutions are sampled. Here, we make use of our choice of the relative weighting of the energy terms $A$ and $B$ (see Tab. 1) in the Ising formulation, which ensures that only feasible solutions are minima in the energy landscape. Therefore, we can detect violations when $E_B > 0$ and search for the spin that causes the largest amount of violations. Subsequently, we change the spin value of the node with the highest number of violations to satisfy the constraint and repeat the process until $E_B = 0$. We observe in our experiments that this post-processing step is typically unnecessary, since only in rare cases violating solutions are sampled.

## A.4 Standardization of the Energy Scale

Since the energy scale of CO problems can vary significantly, a good choice of hyperparameters like the initial temperature $T_0$, learning rate and relative weighting between the policy and value loss can vary between different CO problem instances. Therefore, we standardize the energy scale that makes the choice of good hyperparameters easier. For this purpose we first express the binary energy function $E(q)$ (see. Tab 1) in terms of spins, i.e. as $E(\boldsymbol{\sigma})$ by substituting $q_i$ with $\frac{\sigma_i+1}{2}$. We then sample states for CO problem instances from the training set by using a Random Greedy Algorithm (RGA, see App. A.6.2). Since we use only a few RGA steps (see App. A.6.2), this algorithm performs only slightly better than a completely random algorithm. From these states the mean energy $\hat{E}_{\mathrm{RGA}}$ and the standard deviation $\mathrm{std}(E_{\mathrm{RGA}})$ over the training dataset is computed. Subsequently, we standardize the energy scale by:

$$\hat{E}(\boldsymbol{\sigma}) = \frac{E(\boldsymbol{\sigma}) - \hat{E}_{\mathrm{RGA}}}{\mathrm{std}(E_{\mathrm{RGA}})}. \tag{11}$$

## A.5 Baseline Methods

We compare VAG-CO with our own implementation of a Mean Field Approximation (MFA) and to Erdos Goes Neural (EGN). In both of these approaches we consider the case with and without annealing. In both methods the probability of a state $\sigma$ factorizes into a product of independent Bernoulli probabilities. Therefore, the state probability is given by:

$$p_\theta(\boldsymbol{\sigma}|E) = \prod_{i=1}^{N} p_\theta(\sigma_i|E). \tag{12}$$

While EGN and MFA both use the same probability distribution they differ in how the energy, or in case of annealing the free energy, is calculated. This is described in the following.

### A.5.1 Mean Field Approximation

We estimate the energy by sampling states from the parameterized probability distribution and apply REIN-FORCE with variance reduction to update the network parameters. The network parameter gradients for one CO Problem Instance are then calculated with

$$\Delta\theta(E) = \mathop{\mathbb{E}}_{\boldsymbol{\sigma} \sim p_\theta(\boldsymbol{\sigma}|E)} \left[ (E(\boldsymbol{\sigma}) - b)\nabla_\theta \log p_\theta(\boldsymbol{\sigma}|E) \right],$$

where $b = \mathbb{E}_{\boldsymbol{\sigma} \sim p_\theta(\boldsymbol{\sigma}|E)}[E(\boldsymbol{\sigma}]$. For a batch of CO Problem Instances $\Delta\theta(E)$ is averaged over multiple $E$.

### A.5.2 MFA with Annealing

In MFA-Anneal we estimate the free energy instead and again update the network parameters with REINFORCE with variance reduction as explained above.

### A.5.3 EGN

Since we consider cases, where the energy function is known in Ising formulation (see Sec. 1) the expectation $\mathbb{E}_{\boldsymbol{\sigma} \sim p_\theta(\boldsymbol{\sigma}|E)}[E(\boldsymbol{\sigma}]$ can be written down in closed form by substituting $\sigma_i \rightarrow 2 \cdot p_\theta(\sigma_i|E) - 1$ [Karalias and Loukas, 2020]. Therefore, the loss function for one CO Problem Instance can be computed with

$$L(E) = E(p_\theta(\boldsymbol{\sigma}|E))$$

and the network parameters can be updated directly by computing the gradients of this loss function.

### A.5.4 EGN with Annealing

In case of EGN-Anneal [Sun et al., 2022] the entropy of the parameterized probability distribution is calculated in closed form with the following formula:

$$H(p_\theta(\boldsymbol{\sigma}|E)) = -\sum_{i}^{N} [p_\theta(\boldsymbol{\sigma}_i|E) \cdot \log p_\theta(\boldsymbol{\sigma}_i|E) + (1 - p_\theta(\boldsymbol{\sigma}_i|E)) \cdot \log(1 - p_\theta(\boldsymbol{\sigma}_i|E)]$$

### A.5.5 Conditional Expectation

To obtain good samples from the MFA approach in a deterministic way, we adopt the Conditional Expectation (CE) method [Raghavan, 1988] as described in Karalias and Loukas [2020].

To introduce randomness into the CE-based solution generation procedure we initialize the random node features (which are processed by the GNN) of every node with a random vector with six binary entries that are drawn from a uniform distribution. Unless stated otherwise, we follow the procedure in [Wang and Li, 2023] and report the best CE result of eight different random node feature initializations.

## A.6 Greedy Algorithms

### A.6.1 Degree Based Greedy Algorithm

The Degree Based Greedy (DB-Greedy) algorithm [Wormald, 1995] is a polynomial time algorithm for the MIS problem. The DB-Greedy algorithm works in the following way: At first all nodes are sorted according to their degrees. Then, starting from the smallest degree the node is chosen to be part of the independent set. In the next step this node, its neighbors and their corresponding edges are deleted from the graph. These steps are repeated until the graph is empty.

This algorithm can also be applied to the MVC problem by using the fact that the complement of an independent set is a vertex cover. In other words, nodes in the independent set are excluded from the vertex cover and nodes that are not part of the independent set are included into the vertex cover.

### A.6.2 Random Greedy Algorithm

The Random Greedy Algorithm (RGA) is a general approach that can be utilized for solving a broad range of CO problems that can be mapped to the Ising model. Initially, the algorithm randomly samples spin values with uniform probability. Then, for a fixed number of iterations a spin is randomly selected and its value is changed if it decreases the energy value. We set the number of iterations to $N \cdot n_R$, where $N$ is the number of nodes in the graph and $n_R$ is the number of repetitions per node.

For the purpose of the standardization the energy scale (see. App. A.4), we employ this algorithm with $n_R$ set to one.

## A.7 Study on Sampling Methods

In order to find out, how the best samples can be obtained with VAG-CO we study in Fig. 3 three different sampling methods on the RRG-100 MIS and RB-200 MVC dataset. Here we evaluate $\epsilon^*_{\mathrm{rel}}$ on the test dataset and plot it over the number of samples $n_S$ that are used for each graph in the dataset. We also add MFA and DB-Greedy results, where we show for MFA how the method improves when more solutions are sampled. Since DB-Greedy is a deterministic algorithm, we draw a horizontal line that indicates the solution quality of the algorithm. To draw a relation to results that are presented in Fig. 1 (Left, Middle) we also add a vertical dotted line at $n_S = 8$ samples. For VAG-CO we denote the first method as *sampling* (S), where for each graph $n_S$ solutions are sampled according to the corresponding probability distribution. In the second VAG-CO sampling method called *ordered sampling* (OS), we use for each graph $n_O$ different BFS node orderings and sample $n_S/n_O$ states per graph ordering. Finally, we sample VAG-CO solutions with a method called *ordered greedy* (OG), where we generate solutions greedily for each BFS ordering with $n_O = n_S$. Results in Fig. 3 show that the sampling strategy OG always outperforms all other sampling strategies that we proposed for VAG-CO. Additionally, we see that as we increase $n_O$ in the OS sampling strategy, the performance improves consistently. Remarkably, DB-Greedy exhibits the best solution quality when MFA and VAG-CO are allowed only one sample ($n_S = 1$). However, DB-Greedy is outperformed by VAG-CO OG already with a modest amount of $n_S > 1$ samples.

## A.8 Experimental Details

In this section we provide additional details to experiments that are presented in Sec. 6.

**Parameter Checkpointing.** With VAG-CO we checkpoint over the course of training the parameters that obtain the best $\epsilon^*_{\mathrm{rel}}$ and the best $\hat{\epsilon}_{\mathrm{rel}}$ on the validation set. For testing we always use the checkpoint with the best $\hat{\epsilon}_{\mathrm{rel}}$, except for the results in Fig. 3 when the sampling strategies (S) and (OS) are used. Here we use the checkpoints with the best $\epsilon^*_{\mathrm{rel}}$.

**Hyperparameter Tuning.** For MFA-Anneal, the learning rate and initial temperature are tuned on the validation dataset of Enzymes MIS via a grid search. We considered learning rates ($\mathrm{lr} \in [5 \times 10^{-4}, 1 \times 10^{-4}, 5 \times 10^{-5}]$) and initial temperatures ($T_0 \in [0.5, 0.25, 0.1, 0.05]$). With that, the number of GNN layers ($L \in [6, 8, 12]$) are tuned. Finally, the annealing duration ($N_{\mathrm{anneal}}$) is increased until we observe that longer annealing does not lead

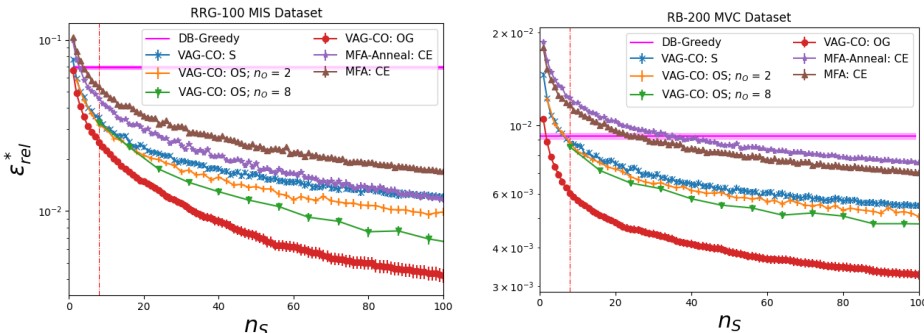

Figure 3: Ablation on different sampling strategies for VAG-CO on the RRG-100 MIS dataset (Left) and on the RB-200 MVC Dataset (Right). The averaged best relative error $\epsilon_{\text{rel}}^*$ on the test dataset is plotted over the number of solutions $n_S$ per graph. Error bars indicate the standard error over the test dataset.

to improvements. For MFA, we tested on ENZYMES MIS the learning rates ($\text{lr} \in [5 \times 10^{-4}, 1 \times 10^{-4}, 5 \times 10^{-5}, 1 \times 10^{-5}]$) and with that tuned the number of GNN layers ($L \in [6, 8, 10, 12]$) and stop training when no significant improvement on the validation set is observed. In MFA and MFA-Anneal, when training on the other datasets we started with the optimal ENZYMES MIS parameters and further tested different learning rates (lr), initial temperatures ($T_0$) and number of GNN layers ($L$) individually on each dataset. For experiments on the RRG-100 MIS and RB-200 MVC we make a more extensive hyperparameter search. In RRG-100 MIS for MFA-Anneal we first tuned the initial temperature in the range of $T_0 \in [0.25, 0.1, 0.05, 0.035, 0.025, 0.015, 0.01]$ and in RB-200 MVC we search for the best initial temperature within the range of $T_0 \in [0.5, 0.25, 0.1, 0.05]$. For MFA and MFA-Anneal, we then iteratively search for the best learning rate in the range of $\text{lr} \in [5 \times 10^{-4}, 1 \times 10^{-4}, 5 \times 10^{-5}]$ and then for the best number of GNN layers $L \in [6, 8, 12]$.

In the experiments with EGN on RB-200 MVC we also performed a extensive hyperparameter seach, where first the best temperature $T_0$ was tuned within the range of $T_0 \in [0.25, 0.1, 0.05]$. Then we tuned the best learning rate within the range of $lr \in [5 \cdot 10^{-4}, 1 \cdot 10^{-1}, 5 \cdot 10^{-5}, 1 \cdot 10^{-5}]$. Then we tuned for the best number of GNN layers withing the range of $L \in [8, 10, 12, 14]$. In EGN without annealing the same hayperparameter search was performed but at $T_0 = 0$. The best hyperparameters of all methods are listed in App. A.13.

On VAG-CO we iteratively tuned the learning rate ($\text{lr} \in [5 \cdot 10^{-4}, 1 \cdot 10^{-3}]$), initial temperature ($T_0 \in [0.05, 0.7, 0.1]$), number of GNN layers ($L \in [3, 4]$) and number of annealing steps ($N_{\text{anneal}} \in [3000, 6000]$) on the ENZYMES MIS validation dataset. We then initially test these hyperparameters on other datasets and adapt the number of annealing steps and the initial temperature. The choice of all hyperparameters in VAG-CO is listed in Tab. 7.

**Ablation on Subgraph Tokenization.** In the ablation on subgraph tokenization in Fig. 1 we keep all hyperparameters the same ,except for the time horizon $\mathcal{T}$ (see Sec. A.9) and the hyperparameter $\lambda$ in the PPO algorithm (see App. A.9). For the subgraph tokenization run we chose $\mathcal{T} = 20$, $\lambda = 0.95$ and $k = 5$. Therefore, the time horizon includes the generation of $\mathcal{T} \cdot k = 100$ spins per graph during the data collection phase (see App. A.9). If we would keep $\mathcal{T}$ the same, when we chose $k = 1$ for the run without subgraph tokenization only $\mathcal{T} \cdot k = 20$ spins would be generated for each graph. Therefore, we use $\mathcal{T} = 100$ when no subgraph tokenization is used. As we always set $\lambda = 1 - \frac{1}{\mathcal{T}}$ this hyperparameter is adapted accordingly.

**Experiments on Random Regular Graphs.** Since GNNs suffer from node ambiguity on Random Regular Graphs (RRGs) Wang and Li [2023] using random node features Abboud et al. [2021] can resolve this issue. Therefore, for the VAG-CO experiment in Fig. 1 (Left) we sample standard Gaussian random node features with the dimension of six and concatenate them to the graph representation of VAG-CO. For MFA and MFA-Anneal random node features are already used (see App. A.5.5) and had not to be added for the RRG experiments.

**Averaged results on hard instances.** In Fig. 1 (Left, Middle) only show results for specific generation parameters $p$ on the RB-200 dataset and for specific values of $d$ on the RRG-100 dataset. Here we report the corresponding averaged results on the RRG-100 MIS and RB-200 MVC dataset in Tab. 6. VAG-CO significantly outperforms all other methods for $n_S = 8$.

## A.9 Proximal Policy Optimization

Proximal Policy Optimization (PPO) [Schulman et al., 2017] is a popular RL algorithm that has two main components. The policy is represented by the network $p_\theta(\sigma_i | G_i)$ with parameters $\theta$. The expected future reward is estimated by the value network $V_\phi(G_i)$ which is parameterized by $\phi$.

| | RB-200 MVC | | RRG-100 MIS | |
|---|---|---|---|---|
| Evaluation metric A.1 | $AR^*$ | (s/graph) | $AR^*$ | (s/graph) |
| DB-Greedy | $1.0088 \pm 0.006$ | 0.01 | 0.931 | 0.003 |
| MFA: CE | $1.0111 \pm 0.0002$ | 0.10 | 0.947 | 0.06 |
| MFA-Anneal: CE | $1.0098 \pm 0.0001$ | 0.10 | 0.955 | 0.06 |
| EGN: CE | $1.012 \pm 0.0001$ | 0.10 | N/A | N/A |
| EGN-Anneal: CE | $1.0087 \pm 0.0002$ | 0.10 | N/A | N/A |
| VAG-CO (ours) | $\mathbf{1.0059 \pm 0.0001}$ | 0.20 | **0.975** | 0.08 |
| Gurobi ($t_{\max} = 0.08$) | $1.0092 \pm 0.0001$ | 0.06 | 0.945 | 0.07 |
| Gurobi ($t_{\max} = 0.1$) | $1.0073 \pm 0.0001$ | 0.07 | 0.954 | 0.09 |
| Gurobi ($t_{\max} = 0.15$) | $1.0052 \pm 0.0001$ | 0.09 | 0.974 | 0.12 |
| Gurobi ($t_{\max} = 0.2$) | $1.0037 \pm 0.0001$ | 0.11 | 0.987 | 0.16 |
| Gurobi ($t_{\max} = 0.3$) | $1.0017 \pm 0.0001$ | 0.13 | 0.995 | 0.22 |
| Gurobi ($t_{\max} = 0.5$) | $1.0006 \pm 0.0001$ | 0.176 | 0.998 | 0.31 |
| Gurobi ($t_{\max} = 1.$) | $1.0001 \pm 0.0001$ | 0.26 | 1.000 | 0.37 |

Table 6: Best approximation ratio $AR^*$ on the test dataset of the RB-200 MVC and RRG-100 MIS dataset. Results where the $AR^*$ is closer to one is better.

**Data Collection.** In PPO the policy is rolled out through the environment in order to collect and store the states, actions, probabilities and the output of the value function into the rollout buffer. The data collection procedure is completed after going through $\mathcal{T}$ steps, where $\mathcal{T}$ is the so-called time horizon. Then, the advantages $A_i := A(\sigma_i, G_i)$ and value targets $V_i^T := V^T(G_i)$ are calculated by making use of the Generalised Advantage Estimation (GAE, Schulman et al. [2016]), where $A_i$ is given by

$$A_i = \delta_i + (\gamma\lambda)\delta_{i+1} + ... + (\gamma\lambda)^{\mathcal{T}-t+1}\delta_{\mathcal{T}-1}, \tag{13}$$

with $\delta_i = R_i + \gamma V(G_{i+1}) - V(G_i)$ and $V_i^T = A_i + V(G_i)$. In our experiments the reward $R_i$ is given by Eq. 6. In our experiments we chose $\gamma = 1.0$ and $\lambda = 1 - \frac{1}{\mathcal{T}}$. We set $\mathcal{T}$ to be approximately equal to the average graph size in the dataset. During the data collection phase we always collect data from $H = 30$ different CO problem instances and for each instance we sample $n_S = 30$ solutions.

**Rollout Buffer.** For the gradient updates in PPO, the loss is estimated with minibatches that are randomly sampled from the rollout buffer. As the rollout buffer contains data of size $H \times n_S \times \mathcal{T}$ we chose to sample data by first sampling $H_{\mathrm{minib}}$ graphs and then for each graph, we chose to sample $N_{\mathrm{minib}}$ solutions and for each solution $S_{\mathrm{minib}}$ time steps are sampled. We report the minibatch settings of each experiment in Tab. 7.

**Training.** As described before, during training minibatches of data $D_{\mathrm{minib}}$ is randomly sampled from the replay buffer. In PPO the overall loss that depends on the $L(\theta, \phi; D_{\mathrm{minib}})$ is given by $L(\theta, \phi; D_{\mathrm{minib}}) = L_1(\theta; D_{\mathrm{minib}}) + c_V L_2(\phi; D_{\mathrm{minib}})$, where the first term depends on the policy and the second term on the value function. To specify a minibatch sample from $D_{\mathrm{minib}}$, we will use the index $n$.

The policy loss for one minibatch sample is then defined as

$$L_1(\theta)_n = -\min\left(I_n(\theta)\,A_n, clip(I_n(\theta), 1-\epsilon, 1+\epsilon)\,A_n\right), \tag{14}$$

where the importance sampling ratio $I_n(\theta) = \frac{p_\theta(\sigma_n|G_n)}{p_{\theta'}(\sigma_n|G_n)}$ is used to compute a weighted Monte Carlo estimate of $A_n$. Here, $\theta'$ represents the old policy parameters that were used to fill the rollout buffer during the data collection phase and $\theta$ are the new parameters that are used in gradient descend. In Eq. 14 the clipping function is used to prevent that the probability of the current policy $p_\theta(\sigma_n|G_n)$ differs by more than a factor $1 \pm \epsilon$ from the old policy $p_{\theta'}(\sigma_n|G_n)$.

Similarly, the value function loss for one sample is given by $L_2(\phi)_n = (V_\phi(G_n) - V_{\phi'}^T(G_n))^2$.

The loss $L(\theta, \phi; D_{\mathrm{minib}})$ is then updated for each minibatch in the rollout buffer so that each sample is used at least once. This procedure is overall repeated $n_{\mathrm{repeat}}$ times, before we increment the epoch step $N_{\mathrm{epoch}}$ and new data is collected with the updated set of parameters. In our experiments we always chose $\epsilon = 0.1$, $c_V = 0.5$ and $n_{\mathrm{repeat}} = 2$.

## A.10 Model Details and GNN Architectures

Before processing the graph representation with GNNs, we encode its node features with an encoder MLP. To obtain node embeddings $\mathcal{N}_j{}^l \in \mathbb{R}^{d(l)}$ with $l \in [1, \ldots, L]$ that depend on the graph structure, the encoded node features are processed by $L$ layers of GNNs. Afterwards, we apply a global sum aggregation to compute a global graph embedding. The global graph embedding is then concatenated to the node embedding $\mathcal{N}_i^L$, where $i$ is the index of the spin whose value is to be generated (see Sec. 3). We then use this embedding to calculate the policy and value function outputs (see App. A.9) with two separate MLPs.

In our experiments, we employ two distinct message passing architectures. The first one is a Message-passing Neural Network (MPNN) [Battaglia et al., 2018] but with linear message functions. The second architecture also includes skip connections (MPNN-skip) [Battaglia et al., 2018] which we use when more than three GNN layers are used.

**Message Passing Neural Network.** Our MPNN layer is defined by the following update for node embeddings:

$$\mathcal{N}_i^{L+1} = \ln \left[ \Psi \left( \mathcal{N}_i^L, \sum_{j \in N(i)} \Phi(\mathcal{N}_j^l, \mathcal{N}_i^l, \mathcal{E}_{ij}) \right) + W_{\text{skip}} \mathcal{N}_i^L \right], \tag{15}$$

where $\Phi(\mathcal{N}_j^l, \mathcal{N}_i^l, \mathcal{E}_{ij})$ is a message update MLP that computes messages with pairwise interactions between nodes. Skip connection are implemented by adding the term $W_{\text{skip}} \mathcal{N}_i^L$ after the node update, where $W_{\text{skip}} \in \mathbb{R}^{d(l+1) \times d(l)}$ is a weight matrix. Finally, layer normalization $\ln[\cdot]$ [Ba et al., 2016] is applied as it is commonly done in Neural Network architectures when skip connections are used [Liu et al., 2021].

In VAG-CO we use a policy and value MLP with three layers. Both MLPs have 120 neurons in the first two layers. The value MLP has only one neuron in the output layer and the policy MLP has $2^k$ output neurons. For the encoder MLP we use a two layer MLP with 40 neurons in each layer. In case of the node update MLP $\Psi$ and the message update MLP $\Phi$ we use 2 layers with 64 neurons in each layer. Layer normalization $\ln[\cdot]$ is applied after every layer within a MLP except in the output layer of the policy and value MLP. We use Rectified Linear Unit (ReLU) activation functions Agarap [2018] except in the output of the policy MLP, where a softmax activation is used and except of the output of the value MLP where no activation is used.

In MFA with and without annealing the model uses an encoder MLP with one layer and 64 neurons. The node update MLP $\Psi$ and the message update MLP $\Phi$ have the same number of neurals as in VAG-CO. The output MLP that is applied on each node after $n$ GNN message passing steps the has three layers with 64 neurons each and the final output layer has 2 neurons with a softmax activation.

### A.10.1 Subgraph Tokenization

For subgraph tokenization we have to make changes to the one-hot encoding as described in Sec. 3 and also to the model architecture as described in App. A.10. The one-hot encoding in the graph representation is adapted so that spins $\sigma_{i:i+k}$ that are going to be generated receive an enumeration that indicates their position in the sliced list of BFS-ordered (see Sec. 3]) indices $i : i + k$. Then, the graph $G_i$ is processed by a GNN that provides a node embedding $GNN_\theta(x_i)$ for each node. Along with a global sum aggregation, the node embeddings of the spins that are going to be generated are then concatenated according to their BFS order (Sec. 3) and further processed by the policy MLP that calculates the probability for each of the $2^k$ spin configurations using a softmax output layer.

When the number of nodes $N$ in the graph is not dividable by $k$, the number vertices in the CO problem instance description has to be increased without changing the inherent optimization objective. This can be realised by adding a sufficient amount of spins into the Ising model (see Sec. 2) with zero spin weight $B$ and no connections to other spins.

### A.11 Training Time and Computational Resources

All runs with VAG-CO were conducted either on A100 Nvidia GPU with 40 GB Memory or an A40 Nvidia GPUs with 48 GB Memory. In case of COLLAB MVC an A100 with 80 GB Memory is used.

The training time of our algorithm depends hyperparameters like the number of annealing steps $N_{\text{anneal}}$ (see App. A.14), the number of edges and nodes of graphs in the dataset, on the GPUs that are used during training, on the time horizon and on the minibatch size that is used for gradient updates in PPO (see App. A.9). For example the TWITTER MVC run with $N_{\text{anneal}} = 4000$, $\mathcal{T} = 30$, $H_{\text{minib}} = N_{\text{minib}} = S_{\text{minib}} = 10$ takes one day, when trained on an A100 40 GB Nvidia GPU. The run on ENZYMES MIS trained on a A40 GPU takes ten hours, when trained with $N_{\text{anneal}} = 6000$, $\mathcal{T} = 20$, $H_{\text{minib}} = N_{\text{minib}} = 15$ and $S_{\text{minib}} = 10$.

## A.12 Time measurement

**Gurobi.** All Gurobi results are conductedon a Intel Xeon Platinum 8168 @ 2.70GHz CPU with 24 cores. All Problems expect for MaxCut are formualted as LIP and we set the number of threads for Gurobi to 26.

**Learned Methods.** For all learned methods (EGN, MFA, VAG-CO) we present the time per graph as if the algorithm are implemented in a parallelized manner, where each sampling process runs on a separate thread. We always report runtimes by using the jit compiled performance of the Graph Neural Network.

## A.13 Hyperparameters

**VAG-CO Hyperparameters.**

All hyperparameters that change across datasets are listed in Tab. 7, whereas hyperparameters that stay the same are specified in the corresponding section (e.g. App. A.10, A.14).

| Dataset | CO Problem Instance | learning rate | $N_{\text{anneal}}$ | $T_0$ | $L$ | $N_{\text{minib}}$ | $H_{\text{minib}}$ | $S_{\text{minib}}$ | $\mathcal{T}$ | $k$ |
|---|---|---|---|---|---|---|---|---|---|---|
| TWITTER | MVC | $1 \times 10^{-3}$ | 4000 | 0.05 | 3 | 10 | 10 | 10 | 30 | 5 |
| COLLAB | MVC | $1 \times 10^{-3}$ | 12000 | 0.05 | 3 | 10 | 10 | 6 | 35 | 5 |
| | MIS | $1 \times 10^{-3}$ | 4000 | 0.05 | 3 | 10 | 10 | 10 | 30 | 5 |
| IMDB-BINARY | MVC | $1 \times 10^{-3}$ | 2000 | 0.05 | 3 | 10 | 10 | 10 | 5 | 5 |
| | MIS | $5 \times 10^{-4}$ | 2000 | 0.05 | 3 | 10 | 10 | 10 | 5 | 5 |
| | MaxCl | $5 \times 10^{-4}$ | 0.03 | 6000 | 3 | 6 | 10 | 7 | 20 | 6 |
| ENZYMES | MIS | $5 \times 10^{-4}$ | 6000 | 0.07 | 4 | 15 | 15 | 15 | 25 | 5 |
| | MaxCl | $5 \times 10^{-4}$ | 6000 | 0.03 | 3 | 6 | 10 | 7 | 20 | 6 |
| PROTEINS | MIS | $5 \times 10^{-4}$ | 10000 | 0.1 | 3 | 15 | 15 | 10 | 35 | 5 |
| MUTAG | MIS | $1 \times 10^{-3}$ | 2000 | 0.05 | 3 | 10 | 10 | 10 | 7 | 5 |
| RB-100 | MVC | $1 \times 10^{-3}$ | 2000 | $5 \cdot 10^{-2}, 5 \cdot 10^{-3}, 0.0$ | 3 | 10 | 10 | 5 | 20 | 5 |
| RB-200 | MVC | $5 \times 10^{-4}$ | 20000 | 0.05 | 3 | 15 | 10 | 10 | 30 | 5 |
| RRG-100 | MIS | $5 \times 10^{-4}$ | 20000 | 0.1 | 4 | 10 | 10 | 10 | 20 | 5 |
| BA 200-300 | MaxCut | $1 \times 10^{-3}$ | 7000 | 0.08 | 4 | 10 | 10 | 10 | 30 | 8 |

Table 7: Hyperparemeters that are used in VAG-CO on different datasets.

**MFA Hyperparameters.**

All runs with MFA used a batch size $H$ of 32 and we sampled $n_S = 30$ solutions. Furthermore, we used 8 GNN layers, and 6 random node features per node. In the RB-200 MVC experiment 10 GNN layers were used. For MFA-Anneal we used a learning rate of $1 \times 10^{-4}$, 2000 annealing steps $N_{\text{anneal}}$ and a start temperature $T_0$ of 0.1 except for RRG-100 MIS and for RB-200 MVC. In RRG-100 MIS a $T_0$ of 0.015 is used and in RB-200 MVC $T_0 = 0.05$. For MFA with and without annealing we trained for 2000 epochs and used a learning rate of $5 \times 10^{-5}$ except for ENZYMS MIS and RB-200 MVC, where a learning rate of $1 \times 10^{-4}$ is used.

**EGN Hyperparameters.**

All runs with EGN on RB-200 MVC have a batch size of $H = 32$. Here, in EGN-Anneal we use 12 GNN layers and in EGN without annealing 10 GNN layers performes best. In EGN-Anneal the initial temperature of $T_0 = 0.1$ with the learning rate of $5 \times 10^{-5}$ obtained the best results. With EGN without annealing the same learning rate is used. Both methods are trained for 2000 epochs.

## A.14 Annealing Schedule

As described in Sec. 3, we change the temperature in the reward Eq. 6 according to a predefined annealing schedule. During the warm-up phase of $N_{\text{warmup}}$ epochs the temperature is held constant at the initial temperature $T_0$. Afterwards the following temperature schedule is applied:

$$T(N_{\text{epoch}}) = T_{\text{anneal}}(N_{\text{epoch}}) \cdot \left[ \cos\left( 2\pi(\lambda + \frac{1}{2}) \frac{N_{\text{epoch}} - N_{\text{warmup}}}{N_{\text{anneal}}} \right) + 1 \right]. \tag{16}$$

Here, $T_{\text{anneal}}(N_{\text{epoch}})$ is the gradually decreasing amplitude of the temperature oscillations:

$$T_{\text{anneal}}(N_{\text{epoch}}) = \frac{T_0}{1 + c \cdot \frac{N_{\text{epoch}} - N_{\text{warmup}}}{N_{\text{anneal}}}}. \tag{17}$$

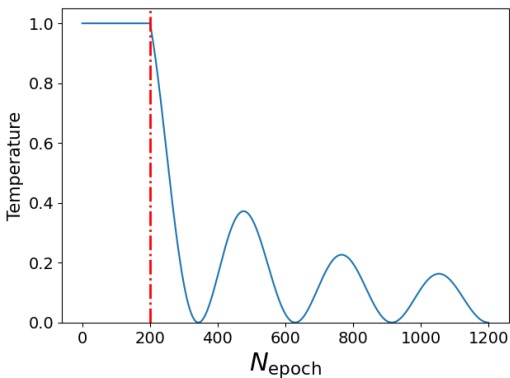

Figure 4: Illustration of the cosine modulated annealing schedule. The plot depicts the annealing schedule with $N_{\text{warmup}} = 200$ and $N_{\text{anneal}} = 1000$ steps. The vertical red line marks the end of the warmup phase.

Here $c$ is a scaling factor that determines the slope of $T_{\text{anneal}}(N_{\text{epoch}})$. The parameter $\lambda$ determines the number temperature oscillations in the schedule and $N_{\text{anneal}}$ is the total number of epochs that follow after the warmup phase. We use $\lambda = 3$, $N_{\text{warmup}} = 400$ and $c = 6$. The course of the annealing schedule is illustrated in Fig. 4. One reason for the usage of cosine modulation function is rather practical, namely that $T = 0$ is reached multiple times during training, which allows an assessment of the training success at an earlier stage for a given set of hyperparmeters. In the absence of cosine modulation, we found it harder to assess the final performance before the entire annealing was finished. Similar periodic schedules for learning rates have been proposed as variants of Stochastic Gradient Descent [Loshchilov and Hutter, 2017].

## A.15  Derivations

### A.15.1  Free-Energy Decomposition into Rewards

In the following we show that using the reward defined in Eq. 6 is consistent with the goal of minimizing the free-energy defined in Eq. 2.
The right-hand side of Eq. 2 contains the expectation of the energy $E(\boldsymbol{\sigma})$ and a term that is proportional to the entropy of $p_\theta(\boldsymbol{\sigma})$. For the energy we obtain the following decomposition into individual steps $i$ of the solution generation process (Sec. 3):

$$E(\boldsymbol{\sigma}) = \sum_{i=1}^{N} \left[ \sum_{j<i} J_{ij}\sigma_j\sigma_i + B_i\sigma_i \right] = \sum_{i=1}^{N} \sigma_i \left[ \sum_{j<i} J_{ij}\sigma_j + B_i \right] = \sum_{i=1}^{N} \Delta E_i \qquad (18)$$

By using an autoregressive factorization the entropy can also be decomposed in the following way:

$$S\big(p_\theta(\boldsymbol{\sigma}|E)\big) = -\sum_{\boldsymbol{\sigma}} p_\theta(\boldsymbol{\sigma}|E) \log p_\theta(\boldsymbol{\sigma}|E) = -\sum_{\boldsymbol{\sigma}} p_\theta(\boldsymbol{\sigma}|E) \left[ \sum_{i=1}^{N} \log p_\theta(\sigma_i|\sigma_{<i}, E) \right]$$
$$= - \mathop{\mathbb{E}}_{\boldsymbol{\sigma} \sim p_\theta} \left[ \sum_{i=1}^{N} \log p_\theta(\sigma_i|\sigma_{<i}, E) \right] \qquad (19)$$

Therefore, we can use this decomposition by using the reward $R_i(G_i, \beta) = -\left[ \Delta E_i + \frac{1}{\beta} \log p(\sigma_i|\sigma_{<i}, E) \right]$. By the relation $F(\theta; \beta, E) = -\mathbb{E}_{\boldsymbol{\sigma} \sim p_\theta}[\sum_i R_i(G_i, \beta)]$, maximizing this reward will then be equivalent to minimizing the free-energy.

### A.15.2  Details on Remark 1

We first restate Corollary 5 of [Dudík et al., 2007] in our notation. In the context of this Corollary our energy function $E$ can be regarded as a single feature and, therefore, we use their result for $n = 1$. In addition, we consider the case in which the samples $\mathcal{S}$ are drawn from the target distribution, i.e. $\pi = p(\beta^*)$. Since the terms Boltzmann distribution and Gibbs distribution can be used interchangeably we obtain:

**Corollary 1** (Corollary 5 of [Dudík et al., 2007]). *Assume that $E$ is bounded in $[0,1]$. Let $\hat{\beta}$ minimize $\mathcal{L}_{\tilde{p}}(\beta) + \lambda|\beta|$ with $\lambda = \sqrt{\ln(2/\delta)/(2m)}$. Then with probability at least $1 - \delta$ for every Boltzmann distribution $p(\beta)$,*

$$\mathcal{L}_{p(\beta*)}(\hat{\beta}) \leqslant \mathcal{L}_{p(\beta*)}(\beta) + \frac{|\beta|}{\sqrt{m}}\sqrt{2\ln(2/\delta)}.$$

Now we consider the result for the case $\beta = \beta*$ and subtract $\mathcal{L}_{p(\beta*)}(\beta*)$. Remark 1 follows by applying the definition of the Kullback-Leibler divergence $D_{KL}$ to the left-hand side.

We note that an equivalent statement, however, with a consideration the Rademacher complexity of $E$ can be derived from Theorem 2 in Cortes et al. [2015].

