# OpenReview forum: "Variational Annealing on Graphs for Combinatorial Optimization"
_NeurIPS.cc/2023/Conference — NeurIPS 2023 poster_

### Official Review · Reviewer_rhwA · 2023-07-03

**Soundness:** 3 good
**Presentation:** 1 poor
**Contribution:** 2 fair
**Rating:** 4
**Confidence:** 4

**Summary:**

This paper uses RL and Annealing to train a learned CO solver. They demonstrate that their algorithm empirically performs well over many problem types and dataset types.

**Strengths:**

1. This paper uses several important baselines.
2. The breadth of the datasets chosen is good.

**Weaknesses:**

1. You are missing a comma in the first sentence.
2. Use \citep to put parentheses around your citations
3. You misspell Reinforce in the table.
4. The experiments seem relatively limited in that comparison is only for two problem types. Moreover, the authors do not compare with Gurobi to get a range on how solvers that are not learned perform on these problems. Also, the time taken by each algorithm is not used to compare these algorithms. Indeed, these metrics are important to see to compare.
5. I do not understand the contribution of this paper. Indeed, it seems that this paper combines annealing from Sun et al. and RL for CO, which is also known. The subgraph solving seems to reflect many classical CO algorithms and is not new. The subgraph tokenization may be novel. In general, I'm a little confused about what the overall contribution of this paper is in the context of the literature.
6. Also, the improvement in the solving ratio seems very small, enough such that hyperparameter tuning your algorithm and not tuning the baselines can make this difference.

I might also change the title of the paper. Currently, it closely resembles "Annealed Training for Combinatorial Optimization on Graphs" by Sun et. al.

**Questions:**

1. During the RL process, how do you compute the reward of an intermediate graph where spins for certain nodes have not been assigned? Do you only compute the reward for nodes that have assigned spins?

Overall, due to the limitations in the experiments, issues with experiments, and lack of contribution, I am voting to reject this paper. However, if the authors can run some further experiments comparing this paper with Gurobi and including the time taken by each algorithm, I believe this paper will be improved. I am flexible, so if the authors adequately address my concerns, I will increase my score accordingly.

**Limitations:**

They have addressed the limitations.

---

> ### Author Rebuttal · Authors · 2023-08-08
>
> We thank the reviewer for the thoughtful and constructive review and would like to respond on various points in the following.
>
> ### Weaknesses:
>
> **(1-3) Latex citation style, comma, and typos:**
> We will update the manuscript accordingly.
>
> **(4)**
>
> **Limited experimental evaluation on only two different problem types:**
> We managed to extend our evaluation by adding benchmarks for Max-Cut and Max-Clique. For Max-Cut we significantly outperform the recent work by Zhang et al. [2023] on the Barabási & Albert (BA) dataset with 200 - 300 nodes. In case of the Max-Clique problem we benchmark on the ENZYMES and IMDB-Binary dataset and significantly improve upon the results in Karalias et al. [2022]. The results are shown in Tab. 4 & 5 in the rebuttal file.
>
> **No comparison to Gurobi:**
> As pointed out in “A.1 Evaluation Metrics” all our metrics are relative to Gurobi solutions (references in L272 and L286 and in Table 2 & 3).
> We will place a corresponding statement at the end of the first paragraph of the “Experiments” section. Importantly, our updated results in Figure 1 left and middle (see rebuttal file) show that VAG-CO outperforms Gurobi for similar time budgets on the hardest synthetic problem settings, i.e. for low values of $d$ and $p$.
>
> **Lack of runtime comparisons:**
> We agree that runtimes are an important aspect and added the corresponding times for all of our experiments. The rebuttal file contains runtimes for Table 2 & 3 and for all remaining experiments in Fig. 1. In Table 2 we currently cannot provide runtimes for the results that were reported from Karalias et al. [2022] since they did not provide this information for their MIS experiments.
> Figure 3 in the rebuttal file shows that Subgraph Tokenization does not only yield higher solution qualities (as already shown in Fig. 1, right) but also sizable runtime reductions.
>
> **(5) Relation to Sun et al. [2022] and RL for CO and unclear contribution and novelty:**
> The reviewer points out that Sun et al. [2022] also use annealing. This is correct, they use annealing in combination with a mean-field method. We cite them in the corresponding section on “Variational Annealing” in L249 (and in L36).
> We are also well aware that RL for CO is not at all new and we do not claim that is new. On the contrary, in the “Related Work” section in L211 we dedicate a subsection to “Unsupervised Learning and Reinforcement Learning” to discuss the relation of our work to RL for CO.
> We would like to re-emphasize our main contributions, including:
>
> (i) identification of a central limitation related to mean-field methods in numerous recent works,
> (ii) introduction of Subgraph Tokenization to enable efficient autoregressive graph generation,
> (iii) state-of-the-art results on several popular CO problems on real-world and on synthetic datasets,
> (iv) a motivation for annealing from statistical learning theory.
>
> With regard to the novelty the reviewer criticizes that “subgraph solving” is not new, which is correct and we do not  intend to claim that this is a novelty in any way. The reviewer acknowledges that Subgraph Tokenization “may be novel”. To the best of our knowledge it is novel. In case the reviewer still has significant remaining doubt we would like to understand the underlying reasons.
>
> **(6) Small improvements, hyperparameters could explain them:**
> The reviewer states that improvements with VAG-CO are rather small and that in such scenarios insufficiently tuning the hyperparameters of baselines could explain superior performance.
> We agree that on real-world graphs (Table 2 & 3) the achievable remaining improvements are indeed small. For this reason we add synthetic problems which are known to be hard. On these problems there is more margin with respect to optimal solutions. We compare MFA methods, VAG-CO, DB-Greedy. In the updated figures we added results for Gurobi with various runtimes and find that we can outperform Gurobi on the hardest problems. The improvements of VAG-CO with respect to the other methods on these problems are sizable and highly significant (Fig. 1, left and middle in the rebuttal file).
> Importantly, we report our hyperparameter optimization strategies in detail in “A.7 Experimental Details” L630 ff. which shows that the hyperparameter search for the MFA methods in Fig. 1, left and middle was actually very extensive and certainly not less extensive than for VAG-CO.
>
> The reviewer criticizes that the title is too similar to the title of Sun et al. [2022]. If it is possible we would follow the reviewer’s advice and change the title to e.g. “Beyond Mean-Field: Autoregressive Graph Generation for Combinatorial Optimization”.
>
> ### Questions:
>
> **(1) How is the reward calculated for intermediate graphs?**
> We calculate the rewards only for the spins that have already assigned values (see Eq.6, L144, and  A.13.1 Free-Energy Decomposition into Rewards).
>
> Based to the main points of this review we improved our work in terms of additional experiments, comparisons to Gurobi, and by adding runtimes. We hope that these improvements will be reflected in the updated score.
>
> Zhang et al. [2023], “Let the Flows Tell: Solving Graph Combinatorial Optimization Problems with GFlowNets”, arXiv:2305.17010
> Sun et al. [2022], “Annealed Training for Combinatorial Optimization on Graphs”, arXiv:2207.11542
> Karalias et al. [2022], “Neural Set Function Extensions: Learning with Discrete Functions in High Dimensions”, arXiv:2208.04055

---

> > ### Comment · Reviewer_rhwA · 2023-08-10
> > **Request for Further Clarification**
> >
> > I want to state I appreciate the detailed rebuttal. I have a few follow-up questions.
> >
> > ### Gurobi
> > On real datasets such as Collab and Twitter, if you give Gurobi the same amount of time as VAG-CO to run, which is the stronger method in terms of approximation ratio? I understand you used Gurobi to calculate the approximation ratio, but which is the faster method on real datasets? I believe this is an important point for me.
> >
> > ### Questions about the experiments
> > What do you mean when you say you can't provide the runtimes of Karalias et al. 2022 since they didn't provide the results? Are you not rerunning their algorithm on your setup? What about the method from Sun et al. 2022? I believe this is MFA-Anneal CE in your paper. Do you rerun their implementation or rely on the numbers from their paper?
> >
> > ### Code for the experiments
> > May I ask why there was no code provided with this submission? I would have liked to check this code for an empirical paper such as this.
> >
> >
> > ### Questions on Experiments
> > Namely, of the four points you mentioned, the problem of MFA relying on the assumption that the parameters are statistically independent is known, so I do not believe point 1 contributes to this paper. Moreover, point 4 was discussed already in Sun et al. 2022. However, I agree that (ii) subgraph tokenization seems to be the central contribution of this paper that no other paper has done. Is there anything else I am missing?

---

> > > ### Comment · Reviewer_fy9s · 2023-08-11
> > >
> > > The code is indeed provided: https://github.com/VAG-CO/VAG-CO

---

> > > ### Author Response · Authors · 2023-08-11
> > > **Further Clarification**
> > >
> > > We thank the reviewer for the prompt reaction. We are happy to read that the novelty and importance of Subgraph Tokenization is now acknowledged by the reviewer. We also point out  that the reviewer is not correct in suggesting that we do not provide code. Our git repository is linked in a footnote on page one.
> > >
> > > **“On real datasets such as Collab and Twitter, if you give Gurobi the same amount of time as VAG-CO to run, which is the stronger method in terms of approximation ratio? ... which is the faster method on real datasets?”**
> > > In our experiments on MVC, MIS, Max-Cut, and Max-Clique Tab. 2,3,4,5 and Fig. 1 in the rebuttal file we report the Gurobi performance for 2-4 different Gurobi runtime limits which were chosen such that at least one runtime limit is comparable to the corresponding runtimes of VAG-CO. In the experiments on real-world graphs (Tab. 2,3,4) Gurobi is often able to solve the problems optimally within runtimes that are shorter than those for learned methods. Thus for these comparably easy CO problem instances Gurobi is the best performing method.
> > > Results on the hard synthetic datasets in Fig. 1 left and middle in the rebuttal file show again that Gurobi is superior for the easier synthetic problems, i.e. in Fig. 1 left the $AR^*$ is better for Gurobi at low $d$ and in Fig. 1 middle $AR^*$ is better for Gurobi at high $p$. For the particularly hard problems (high $d$, low $p$) VAG-CO outperforms or matches the results of Gurobi for comparable time budgets.
> > >
> > > **“What do you mean when you say you can't provide the runtimes of Karalias et al. 2022 since they didn't provide the results? Are you not rerunning their algorithm on your setup?”**
> > > As we state in the captions of Tab. 2 - 5 “(r)” indicates that these results are taken from the corresponding references - this does also hold for the corresponding runtimes. That means indeed that we did not run the methods from Karalias et al. [2022] ourselves but that we only report these results. We want to point out here that re-running all competing methods ourselves is computationally infeasible for us.
> > > **“What about the method from Sun et al. 2022? I believe this is MFA-Anneal CE in your paper.”**
> > > No, this is not “MFA-Anneal CE” in our paper. In contrast to Sun et al. [2022] our “MFA-Anneal CE” is trained via REINFORCE as stated in L273: “We also report results of our own implementation of an MFA-based method that is trained with REINFORCE.”.
> > > **“Do you rerun their implementation or rely on the numbers from their paper?”**
> > > As pointed out above “MFA-Anneal CE” is not the method of Sun et al. [2022]. We ran all experiments that are not indicated as “reported” by “(r)” by ourselves. This includes “MFA-Anneal CE”.
> > >
> > > The reviewer criticizes the four contributions of this paper that we listed in the response to the reviewer’s initial review.
> > > **“Namely, of the four points you mentioned, the problem of MFA relying on the assumption that the parameters are statistically independent is known, so I do not believe point 1 contributes to this paper.”**
> > > Our first point reads: “identification of a central limitation related to mean-field methods in numerous recent works”.
> > > The reviewer correctly states that the mean-field approximation (MFA) relies on the assumption that the parameters are statistically independent. In fact, that is the defining property of MFA. But this is not the point of our first contribution.
> > > Our contribution is that we point out that many recent methods in the field of neural CO rely on MFA and are limited by this assumption. We show empirically that by using a more expressive approach that does not rely on MFA we can obtain superior performance in particular on CO problem instances that are hard.
> > > In case the author is aware of any prior work that made this point we would be very interested in the corresponding references.
> > > **“Moreover, point 4 was discussed already in Sun et al. 2022.”**
> > > Our fourth point reads: “a motivation for annealing from statistical learning theory”.
> > > This point refers to Remark 1 in L194 in which we make a formal statement about the sample complexity of approximating Boltzmann distributions. We consider this theoretical insight as one of our contributions. We are not aware of any formal learning theoretical statement in Sun et al. [2022].
> > > Based on our Remark 1 we argue that annealing can be regarded “as a principled curriculum learning approach”. The connection between annealing and curriculum was conjectured without learning theoretical arguments in Sun et al. [2022]. For this reason we will cite Sun et al. [2022] as follows in L201:
> > > “A connection between annealing and curriculum learning was put forward less formally already in Sun et al. [2022].”
> > >
> > > Karalias et al. [2022], “Neural Set Function Extensions: Learning with Discrete Functions in High Dimensions”, arXiv:2208.04055
> > > Sun et al. [2022], “Annealed Training for Combinatorial Optimization on Graphs”, arXiv:2207.11542

---

> > > > ### Comment · Reviewer_rhwA · 2023-08-11
> > > > **A few clarifications**
> > > >
> > > > I wanted to confirm a few things:
> > > > 1. Gurobi beats VAG-CO on all datasets besides challenging RB problems, correct?
> > > > 2. Do you have experiments for Karialas et al.'s work on RB problems? I believe you sampled these problems yourself. I have run their methods before. I disagree it is computationally infeasible. I see this work mainly as an empirical paper, so I am trying to understand better where and what this algorithm beats. It consistently beat the greedy baselines, MFA-anneal, and Karalias on real datasets (but not by much). On complex problems, it beats them slightly more but not that much better, having an AR difference of around .005 with the second-best method, and Karalias isn't used there, which is essential. It does not beat Gurobi at all on anything besides on hard problem case, as you mentioned. Therefore, I struggle to see how this, as an empirical paper, should be accepted if the improvement is marginal on most problem types (including the RB model) and it doesn't beat Gurobi, which is a classic benchline to beat in this literature. You also don't use Karalias as a baseline on RB model problems, the problem set where this paper has the most edge. I believe Karialias et al. or Sun et al. to be important baselines here due to their performance on RB problems. Have I misunderstood anything here? I hate to be harsh or discouraging to the authors, and I hope that my feedback is constructive.

---

> > > > > ### Author Response · Authors · 2023-08-16
> > > > >
> > > > > **“1. Gurobi beats VAG-CO on all datasets besides challenging RB problems, correct?”**
> > > > > The reviewer’s question is based on an interpretation of our results that is not valid for the following reason: the extent of the parallelization of the solution search in both methods is arbitrary and thus not comparable (Gurobi : 24 CPU cores, 26 threads; VAG-CO: 1 GPU , 8 threads).
> > > > > Our paper’s contribution is not to decide whether Gurobi or VAG-CO is the better algorithm. We agree that this question is interesting but it is not what our experiments were designed for.
> > > > > Our experiments are designed for the comparison to the state of the art in neural CO. The vast majority of recent neural CO works like e.g. Karalias et al. [2020, 2022] and Sun et al. [2022] use Gurobi to obtain ground-truth and some of them report the corresponding runtimes but they do not interpret their results as a comparison between the general quality of their methods and Gurobi. Further examples are Wang et al. [2023] and Böther et al. [2022] where the authors do occasionally compare to Gurobi in some experiments. However, they do NOT interpret this as an insight on which algorithm is better in general. Similarly, we refrain from drawing such conclusions in our case which is analogous.
> > > > > We hope that the reviewer will base the review on our actual contributions and not on a problematic re-interpretation of our results.
> > > > >
> > > > > **“2. Do you have experiments for Karialas et al.'s work on RB problems?” &
> > > > > “… and Karalias isn't used there, which is essential.” & “You also don't use Karalias as a baseline on RB model problems … Karialias et al. or Sun et al. to be important baselines … ”**
> > > > > We hope the reviewer appreciates the extension of our results that was prompted by the initial review. After we delivered these results which confirm the strong performance of VAG-CO the reviewer now demands further new experiments. Since we are fully committed to convincing the reviewer we ran experiments on RB for Karalias et al. [2020] (EGN) and Sun et al. [2022] (EGN-Anneal). The results show that the former method performs similar to our MFA and the latter is similar to DB-Greedy (Fig. 1, middle). Importantly, both are clearly outperformed by VAG-CO.
> > > > >
> > > > > | p                | 0.25   | 0.333  | 0.417  | 0.5    | 0.583  | 0.667  | 0.75   | 0.833  | 0.917  | 1.0   |
> > > > > |------------------|--------|--------|--------|--------|--------|--------|--------|--------|--------|-------|
> > > > > | EGN - AR*        | 1.0167 | 1.0171 | 1.0166 | 1.0122 | 1.0115 | 1.0133 | 1.0116 | 1.0105 | 1.0066 | 1.000 |
> > > > > | EGN-Anneal - AR* | 1.0147 | 1.0146 | 1.0123 | 1.0108 | 1.0088 | 1.0095 | 1.0083 | 1.0073 | 1.0042 | 1.000 |
> > > > > | VAG-CO - AR*     | 1.0110 | 1.0102 | 1.01   | 1.0077 | 1.0056 | 1.0063 | 1.0051 | 1.0044 | 1.0015 | 1.000 |
> > > > >
> > > > > **“It consistently beat … but not by much. … , it beats them slightly more but not that much better, having an AR difference of around .005 with the second-best method”**
> > > > > Using absolute differences between relative metrics is mis-leading. Even the worst possible MVC assignment on RB-200 would only result in an $AR^*$ optimality gap of .114. Likewise, an improvement of .005 in $AR^*$ represents a reduction of the optimality gap by 32% with respect to the second best method (DB-Greedy/EGN-Anneal). This is by no means a minor improvement.
> > > > >
> > > > > **“It does not beat Gurobi at all on anything besides on hard problem case, as you mentioned.”**
> > > > > We commented on the reviewer’s mis-conception in this point in the answer to “1. Gurobi beats…”.
> > > > >
> > > > > **"Therefore, I struggle to see how this, as an empirical paper, should be accepted if the improvement is marginal on most problem types (including the RB model) and it doesn't beat Gurobi, which is a classic benchline to beat in this literature.”**
> > > > > We discuss the reviewer’s mis-conception in judging the performance gap of VAG-CO in our answer to the statement “It consistently…”.
> > > > > Concerning the problems with the question of whether we “beat” Gurobi, please refer to our answer to “1. Gurobi beats…”. This answer will also clarify that claims of superiority or inferiority of neural CO methods with respect to Gurobi are NOT common in neural CO. Therefore, even IF our results would support the claim that VAG-CO was worse than Gurobi, it would be mis-guided to criticize our work for that. None of the relevant previous works made general claims on their method being a superior algorithm to Gurobi and this question is not directly related to any contribution of our work.
> > > > >
> > > > > Karalias et al. [2020], “Erdos Goes Neural: an Unsupervised Learning Framework for Combinatorial Optimization on Graphs”, arXiv:2006.10643
> > > > > Karalias et al. [2022], “Neural Set Function Extensions: Learning with Discrete Functions in High Dimensions”, arXiv:2208.04055
> > > > > Sun et al. [2022], “Annealed Training for Combinatorial Optimization on Graphs”, arXiv:2207.11542
> > > > > Böther et al. [2022], “What's Wrong with Deep Learning in Tree Search for Combinatorial Optimization”, arXiv:2201.10494

---

> > > > > > ### Comment · Reviewer_rhwA · 2023-08-17
> > > > > > **Updated Score**
> > > > > >
> > > > > > Dear Authors,
> > > > > > Thank you for your update. Your results on the RB model over Sun's and Karalias's methods are good to have. While it is my opinion that the empirical benefit is not strong and the variance could have accounted for the minute increases in most cases, the consistent increase over several domains is strong. In my experience, the differences (even given you saying it is 32% of an increase) can be accounted for in the standard deviation of the results. I also have issues with the presentation. I have improved my score to borderline reject and I believe that is fair given my grievances. Thank you.

---

> > > > > > > ### Author Response · Authors · 2023-08-21
> > > > > > >
> > > > > > > **“the empirical benefit is not strong and the variance could have accounted for the minute increases in most cases”**
> > > > > > > The standard errors over 3 seeds are reported in Tables 2,3, and 4 in the rebuttal file for the real-world datasets. For the $AR^*$ results we utilize the Conditional Expectation (CE) procedure for sampling the learned models for mean-field methods. VAG-CO clearly outperforms these methods just by sampling without requiring any further non-learned algorithmic component like CE. In particular, VAG-CO is significantly the single best method on three data sets. On additional four datasets it is among the best performing methods with one or two other methods. VAG-CO is significantly outperformed on only one dataset. On two datasets more than three methods achieve an $AR^*$ of 1.0 and VAG-CO is always among them. This is a remarkable advancement for this field since it shows that using a sufficiently expressive ansatz mitigates the need for hard-coded heuristics like CE.
> > > > > > > This aspect is substantiated by the $\hat{AR}$ results which allow us to assess directly the quality of the learned distribution over the solution space. Here VAG-CO is compared to the two strongest mean-field methods (MFA and MFA-Anneal) without CE. VAG-CO is significantly the single best method on eight datasets and is outperformed only once. On one dataset we observed no significant difference between the studied methods.
> > > > > > > Overall, VAG-CO is clearly the best performing method in these real-world dataset benchmarks from the recent literature.
> > > > > > >
> > > > > > > **“the differences (even given you saying it is 32% of an increase) can be accounted for in the standard deviation of the results”**
> > > > > > > In the following we report the standard errors for the RB results with respect to the solution generation procedure. For all methods we repeated the procedure to obtain $AR^*$ 12-times. For VAG-CO we repeatedly sampled new solutions and for the EGN-based methods we introduce randomness by utilizing random node-features as done in Karalias et al. [2020] and Wang et al. [2023]. The following table reports the average $AR^*$ for the 12 sample generation procedures along with the corresponding standard errors. The large performance advantage (lower is better) of the VAG-CO model is highly significant, in particular, for hard tasks (low $p$).
> > > > > > > In accordance with the real-world benchmarks, these computationally extensive experiments on hard synthetic CO problems clearly underline that VAG-CO represents an improvement with respect to the state of the art in neural CO.
> > > > > > >
> > > > > > > | $p$                | 0.25                | 0.333               | 0.417               | 0.5                 | 0.583               | 0.667               | 0.75                | 0.833               | 0.917               | 1.0            |
> > > > > > > |------------------|---------------------|---------------------|---------------------|---------------------|---------------------|---------------------|---------------------|---------------------|---------------------|----------------|
> > > > > > > | EGN - $AR^*$        | 1.01771 $\pm$ 0.00010 | 1.01693 $\pm$ 0.00007 | 1.01657 $\pm$ 0.00011 | 1.01278 $\pm$ 0.00010 | 1.01187 $\pm$ 0.00010 | 1.01267 $\pm$ 0.00007 | 1.01216 $\pm$ 0.00009 | 1.01088 $\pm$ 0.00011 | 1.00713 $\pm$ 0.00007 | 1.0000 $\pm$ 0.0 |
> > > > > > > | EGN-Anneal - $AR^*$ | 1.01480 $\pm$ 0.00006 | 1.01415 $\pm$ 0.00005 | 1.01241 $\pm$ 0.00008 | 1.01028 $\pm$ 0.00010 | 1.00892 $\pm$ 0.00005 | 1.00966 $\pm$ 0.00010 | 1.00858 $\pm$ 0.00005 | 1.00715 $\pm$ 0.00007 | 1.00411 $\pm$ 0.00007 | 1.0000 $\pm$ 0.0 |
> > > > > > > | VAG-CO - $AR^*$     | 1.01134 $\pm$ 0.00007 | 1.01037 $\pm$ 0.00006 | 1.00941 $\pm$ 0.00007 | 1.00736 $\pm$ 0.00006 | 1.00534 $\pm$ 0.00011 | 1.00551 $\pm$ 0.00009 | 1.00509 $\pm$ 0.00005 | 1.00438 $\pm$ 0.00009 | 1.00152 $\pm$ 0.00005 | 1.0000 $\pm$ 0.0 |
> > > > > > >
> > > > > > > **“I also have issues with the presentation.”**
> > > > > > > As we have demonstrated multiple times above, we are absolutely willing to adapt our work according to the reviewer’s critique. In this case, however, the critique is too vague.
> > > > > > >
> > > > > > > Finally, we thank the reviewer for the continuous engagement. We are happy that the reviewer acknowledges that: "the consistent increase over several domains is strong". We hope that our response above eliminates the reviewer’s remaining doubt on the significance of our experimental results in the context of the literature.
> > > > > > > We stress that besides these strong experimental results, our work encompasses the introduction and demonstration of the graph generation technique Subgraph Tokenization, a statistical learning-theory perspective on variational annealing, and the identification of a limitation inherent to the application of the mean-field ansatz in the recent neural CO literature.
> > > > > > >
> > > > > > >
> > > > > > > Karalias et al. [2020], “Erdos Goes Neural: an Unsupervised Learning Framework for Combinatorial Optimization on Graphs”, arXiv:2006.10643
> > > > > > > Wang et al. [2023], “Unsupervised Learning for Combinatorial Optimization Needs Meta-Learning”, arXiv:2301.0311

---

### Official Review · Reviewer_59Wh · 2023-07-05

**Soundness:** 3 good
**Presentation:** 3 good
**Contribution:** 3 good
**Rating:** 6
**Confidence:** 3

**Summary:**

The authors present Variational Annealing on Graphs for Combinatorial Optimization (VAG-CO), a novel method for tackling combinatorial optimization problems. This method presumably combines elements of variational inference, annealing, and graph theory to form an effective optimization approach.

=====
After reading the rebuttal, the authors have convinced me to increase my score.

**Strengths:**

Novel Methodology: The paper presents a new approach to solve combinatorial optimization problems, Variational Annealing on Graphs for Combinatorial Optimization (VAG-CO), which is innovative and adds value to the existing literature.

Overcoming Limitations of Existing Methods: The authors identify the limitations of the widely used Mean-Field Approximation (MFA) and propose a way to overcome these, which demonstrates a deep understanding of the problem space.

Improvement in Efficiency: The use of sub-graph tokenization and entropy regularization to improve the efficiency of the training and inference process could potentially revolutionize the way these types of problems are solved.

**Weaknesses:**

1. The experiments are only completed on synthetic data. It will be more convincing if the proposed method can be applied and compared on some real data.
2. Maximum Independent Set (MIS) and Minimum Vertex Cover (MVC) are well-known problems in graph theory and are widely used as benchmark problems in combinatorial optimization. However, they only represent a subset of the vast array of combinatorial optimization problems. A more extensive evaluation of the proposed method would involve its application to a larger and more diverse set of problems. This would enable a more comprehensive understanding of its capabilities and limitations. If the authors want to claim the wide application of the proposed method, they may want to conduct experiments for Traveling Salesman Problem, Knapsack Problem, Vehicle Routing Problem etc.

**Questions:**

How do the authors define hard problems in the paper? What are the criteria?


**Limitations:**

The experiments are not extensive.

---

> ### Author Rebuttal · Authors · 2023-08-08
>
> We thank the reviewer for the thoughtful and constructive review. We are happy to read that:
> - our method "is innovative and adds value to the existing literature",
> - we "identify the limitations of the widely used Mean-Field Approximation (MFA) and propose a way to overcome these, which demonstrates a deep understanding of the problem space.",
> - the "The use of sub-graph tokenization and entropy regularization ... could potentially revolutionize the way these types of problems are solved.".
>
> Please find comments to the review in the following.
>
> ### Weaknesses:
>
> **(1): “The experiments are only completed on synthetic data. It will be more convincing if the proposed method can be applied and compared on some real data.”**
> We are not sure about the reviewer’s definition of “synthetic”. Many of the graph datasets that we use like TWITTER, PROTEINS, COLLAB, IMDB-Binary, MUTAG, and ENZYMES are considered to be real-world datasets in the literature.
>
> **(2): “A more extensive evaluation of the proposed method would involve its application to a larger and more diverse set of problems.“**
> We managed to extend our experimental evaluation by adding benchmarks for MaxCut and MaxClique. For MaxCut we significantly outperform the recent work by Zhang et al. [2023] on Barabási & Albert (BA) graphs with 200 - 300 nodes. In case of the MaxClique problem we benchmark on the ENZYMES and IMDB-Binary dataset and significantly improve upon the corresponding results in Karalias et al. [2022]. These results are shown in Tab. 4 and 5 in the rebuttal file.
>
> ### Questions:
> **(1): “How do the authors define hard problems in the paper? What are the criteria?”**
> Our claim of hardness of MIS on Random Regular Graphs (RRGs) is based on the observation of a clustering transition for degrees $d > 16$ in the corresponding independent sets (see Barbier et al. [2013] for details). In the context of neural CO it was argued in Angelini et al. [2022] that these MIS instances are hard. They write:
> “We have argued that for the MIS using $d$-RRG with $d < 16$ is likely to be an easy problem and the test would be not very selective (we have in mind now only smart algorithms, not the GNN of Ref. [4] whose performances are so poor to be rejected even by easy tests). However, for larger $d$ we expect the optimization to become much more demanding because the clustering of the IS of large size is likely to create relevant barriers that affect any algorithm searching for the MIS.”. This claim is empirically compatible with our updated Fig. 1 left in the rebuttal file. It shows that all compared algorithms become worse as $d$ is increased from 3 to 20. Remarkably, if Gurobi is restricted to the same runtime as VAG-CO, i.e. 0.08 s (or even to a longer runtime of 0.1 s), it is outperformed by VAG-CO for $d > 3$.
> The hardness of the RB graphs for the MVC problem is based on a correspondence between the MVC and forced satisfiable SAT problems on RB graphs (see Xu [2004] for details). These graphs were also used recently in Wang et al. [2023]. In agreement with their observations we observe in Fig. 1 middle in the rebuttal file that for low values of the parameter $p$ all methods, including time-limited Gurobi, exhibit a rise in the approximation ratio, i.e. these problems become harder for all investigated methods.
>
> Barbier et al. [2013], “The hard-core model on random graphs revisited”, arXiv:1306.4121
> Angelini et al. [2022], “Cracking nuts with a sledgehammer: when modern graph neural networks do worse than classical greedy algorithms”, arXiv:2206.13211
> Wang et al. [2023], “Unsupervised Learning for Combinatorial Optimization Needs Meta-Learning”, arXiv:2301.0311
> Xu [2004], "BHOSLIB: Benchmarks with Hidden Optimum Solutions for Graph Problems", http://vlsicad.eecs.umich.edu/BK/Slots/cache/www.nlsde.buaa.edu.cn/~kexu/benchmarks/graph-benchmarks.htm

---

### Official Review · Reviewer_fy9s · 2023-07-05

**Soundness:** 2 fair
**Presentation:** 2 fair
**Contribution:** 3 good
**Rating:** 6
**Confidence:** 3

**Summary:**

This paper explores the use of autoregressive deep generative models for solving combinatorial problems.

Starting from an optimization problem, the authors reformulate it using a Boltzmann distribution on the solution set.
This distribution can then be approximated with a parametric family, typically a neural network, following a variational approach (minimization of the free energy).

While many previous works use the independent (or "mean field") approximation, it is suggested that building the solution iteratively is more expressive, and thus performs better.
In this case, the autoregressive model used is a graph neural network, which gives a conditional probability for the current spin given past choices.
To increase inference speed, two tricks are introduced: subgraph tokenization (making several decisions at once) and dynamic graph pruning (removing fixed vertices from the graph).

Training is achieved using a temperature annealing scheme which favors gradual concentration around global optima.
This annealing process enjoys an interpretation in terms of sample complexity and curriculum learning.

Numerical experiments on the maximum independent set and the minimum vertex cover problem show the promise of this new method.

**Strengths:**

### Originality

This paper describes a new combination of known ideas, coherently motivated by the specific goal of solving hard optimization problems:

- formulation of optimization in the language of statistical physics
- variational approximation of the Boltzmann distribution
- iterative construction of a solution
- RL framework with partial rewards
- temperature annealing

As far as I can tell, the training tricks related to tokenization and pruning seem to be novel.

### Quality

The proposed model makes theoretical sense, and rests on solid foundations of previous work.
I particularly enjoyed the interpretation of annealing as curriculum learning: as the temperature gets lower, the learning gets harder.

As for the numerical experiments, they appear to be thorough and include a wide variety of algorithms.
The authors also took care to test on harder instances whenever nearly optimal solutions were too easy to reach.
They do however exhibit a few inconsistencies, which I point out below.

### Clarity

n.a.

### Significance

This paper contributes to a wider discussion on the best variational ansatz for combinatorial optimization.
Comparing mean field with autoregression is an important first step, and hopefully the field can evolve towards more generic structured representations.

Subgraph tokenization is also an interesting idea, which deserves to be pushed further.

**Weaknesses:**

### Originality

### Quality

A questionable algorithmic choice is the arbitrary order in which nodes are processed.
It isn't obvious why BFS ordering of the vertices makes more sense than other options.
The same goes for subgraph tokenization: why take the $k$ next vertices in the BFS order, even though they might be very far apart from each other?

As for the experiments, the realistic datasets of the first batch are seemingly too easy, since every method implemented by the authors (including the baseline DB-Greedy) shows near-perfect accuracy.
The results of Table 2 are very surprising to me, as both EGN and MFA are mean field methods with conditional expectation decoding, yet they exhibit wildly different performances on the minimum AR.
On the other hand, the average AR is only reported for new methods, not for the state of the art.
I would welcome more detail from the authors on this point.
In any case, the benefits of autoregression as opposed to independence are not definitively proven by this series of benchmarks.

The synthetic datasets of the second batch are designed to be harder to solve, but the associated plots do not include other methods from the state of the art, which were analyzed on realistic datasets only.
Again, clarification from the authors would be very welcome.

### Clarity

The notations are sometimes a bit hard to follow.
Several algorithms included in the benchmark suite are only mentioned earlier in the text.

### Significance

Depending on the validity and fairness of the benchmarks, the results may be less significant than announced by the authors.
I look forward to the rebuttal period to enlighten this aspect.

**Questions:**

L121: Why BFS in particular? Is there a way to make the order itself parametric?

L160: Is there a way to perform subgraph tokenization that does not scale exponentially with $k$?

L198: Are there other insights from curriculum learning that we could draw inspiration from? I'm not at all familiar with this literature

L215: What if our problem provides no natural way to define partial rewards?

L312: Why change the evaluation metric from approximation ratio to relative error? Not sure I grasp the difference

**Limitations:**

The authors do not discuss many limitations of their work, although they mention the additional complexity.

Societal impact is irrelevant here.

---

> ### Author Rebuttal · Authors · 2023-08-08
>
> We thank the reviewer for the thoughtful and constructive review and would like to respond on various points mentioned in the weaknesses and questions.
>
> ### Quality:
>
> **BFS:**
> See answer to question on L121 below.
>
> **“the realistic datasets of the first batch are seemingly too easy”:**
> These are datasets that are used in recent works in this field. We agree with the reviewer that these datasets might be too easy (see L304). For this reason we decided to benchmark also on synthetic datasets that are known to yield hard MIS and MVC problems.
>
> **“The results of Table 2 are very surprising to me, as both EGN and MFA are mean field methods with conditional expectation decoding, yet they exhibit wildly different performances on the minimum AR.”**
> A possible reason might be the training method: in EGN the gradients are calculated directly from exact expectation values while our MFA estimates gradients via REINFORCE. The latter approach might be less prone to getting stuck in local minima due to an increased variance in the gradient estimation.
> Interestingly, the recent work of Zhang et al. [2023] also states in App. D that EGN is worse than any of their baselines on their MIS benchmarks.
>
> **“On the other hand, the average AR is only reported for new methods, not for the state of the art.”**
> The purpose of the average AR metric in Table 2 & 3 is to compare the best achievable solution quality for VAG-CO with mean-field based approaches when they do not use conditional expectation. Hence, these results can be regarded as an ablation study in which no non-learned decoding method is used but where we simply report the average AR of the sampled solutions. The computational requirements of including all other under-performing (in terms of $AR^*$) methods in this ablation study would be sizeable. Consequently, we decided to consider only the best performing mean-field methods.
>
> **"The synthetic datasets of the second batch are designed to be harder to solve, but the associated plots do not include other methods from the state of the art…."**
> Since these experiments are computationally expensive we conducted them only for the best performing methods in terms of $AR^*$ in Table 2 & 3. However, due to a request by reviewer rhwA we added the results on synthetic dataset for Gurobi with several time limits to the rebuttal file (Fig. 1).
>
> ### Clarity:
>
> **“The notations are sometimes a bit hard to follow. Several algorithms included in the benchmark suite are only mentioned earlier in the text.”**
> If specific weaknesses of the notation are pointed out to us we would be happy to improve them.
> We will introduce each algorithm in the experiment section of the updated manuscript.
>
> ### Questions:
>
> **L121: Why BFS in particular?**
> BFS will typically order the nodes such that the next $k$ spins are likely to be directly connected to already generated spins and will typically have several generated spins in their neighborhood. Consequently, the newly generated spins receive direct information on the already generated spins via message passing. For example, with depth-first search (DFS) one would expect that the newly generated spins have fewer already generated spins in the neighborhood. Consequently, in DFS there would be less information flow from the already generated spins to the spins that are to be generated next. This should lead to an increased probability of generating tokens that are sub-optimal once they become connected to the previously generated spins.
> Investigating this question empirically could be an interesting direction for future work.
>
> **L121: Is there a way to make the order itself parametric?**
> One can certainly make the order of the spins parametric, e.g. by sampling the order of spins from a probability distribution that is obtained via attention. This would be an interesting direction for future work.
>
> **L160: Is there a way to perform subgraph tokenization that does not scale exponentially with k?**
> One can reduce the number of possible tokens by masking out tokens that violate constraints of the CO problem (e.g. the independence condition in MIS). Whether this would result in a sub-exponential scaling depends on the specific CO problem instance.
>
> **L198: Are there other insights from curriculum learning that we could draw inspiration from?**
> Yes, one could increase the size of the graphs throughout training like in Lisicki et al. [2020].
> More generally, one could cast the task of generating suitable training graphs for curriculum learning in neural CO as a task for a separate agent like in “Teacher-Student Curriculum Learning” by Matiisen et al. [2017]. Here a teacher learns to generate the tasks (e.g. graphs in CO) for a student that learns to solve the corresponding CO problem.
>
> **L215: What if our problem provides no natural way to define partial rewards?**
> Without partial rewards the RL setting would be changed to an MDP with sparse rewards that are received once a complete solution was generated. This RL problem would be harder but our method could still be applicable.
>
> **L312: Why change the evaluation metric from approximation ratio to relative error?**
> The reason was to use in all three subplots of Fig. 1 metrics where lower is better. For the approx. ratio (AR) lower is better for MVC while for MIS higher is better. We agree with the reviewer that it might be better to not use the relative error in Fig. 1 (left) and to stick to $AR^*$. We will do so in the updated manuscript. The updated Fig. 1 can be found in the rebuttal file.
>
>
> Zhang et al. [2023], “Let the Flows Tell: Solving Graph Combinatorial Optimization Problems with GFlowNets”, arXiv: 2305.17010
> Lisicki et al. [2020], “Evaluating Curriculum Learning Strategies in Neural Combinatorial Optimization”, arXiv: 2011.06188
> Matiisen et al. [2017], “Teacher-Student Curriculum Learning”, arXiv: 1707.00183

---

> > ### Comment · Reviewer_fy9s · 2023-08-11
> >
> > Thank you for your answer, which alleviates many of my concerns. In particular, I now understand the numerical experiments a bit better.
> >
> > A few minor details:
> > - The comparison between EGN and your MFA might deserve a remark.
> > - Among the slightly confusing notations: $\omega$ as a probability distribution, $\nu_i$ for the graph nodes (instead of just $i$)
> > - On each result plot, please specify whether lower is higher or worse, since it now changes
> >
> > > BFS will typically order the nodes such that the next spins are likely to be directly connected to already generated spins and will typically have several generated spins in their neighborhood.
> >
> > This might be true at first, but as you expand the BFS, the radius gets larger and there is no guarantee that the $i+1$-th node visited comes from the same parent as the $i$-th one. In fact, it might be located in a very different part of the graph.
> > Plus the whole procedure is very dependent on the choice of source. I agree that DFS seems worse but are there other algorithms one might consider?

---

> > > ### Author Response · Authors · 2023-08-18
> > > **Answer to Comment**
> > >
> > > We thank the reviewer for his prompt response and his follow up suggestions.
> > > ### Regarding the minor details:
> > > **"The comparison between EGN and your MFA might deserve a remark."** \
> > > We will add a remark to the updated version of our manuscript.
> > >
> > > **"Among the slightly confusing notations: $\omega$
> > >  as a probability distribution, $\nu_i$
> > >  for the graph nodes (instead of just $i$
> > > )"** \
> > > We agree that using $\omega$ as a probability distribution may be confusing. Therefore, we will use $q$ instead and clarify in the text that it denotes a probability distribution. Also, we will use $i$ instead of $ \nu_i$ for the graph nodes.
> > >
> > > **"On each result plot, please specify whether lower is higher or worse, since it now changes"** \
> > > We will specify in each figure whether lower or higher is better.
> > >
> > > ### Regarding the BFS question:
> > > We agree with the reviewer that in BFS, when the radius gets large, the next $k$ spins will very likely not be close to each other. However, we do not see why this should be a problem. It would,indeed, be interesting to investigate this in dedicated experiments.
> > >
> > > **"I agree that DFS seems worse but are there other algorithms one might consider?"**\
> > > If one wants to have the property that the next $k$ spins have to be close to each other and that already generated spins are also close to these spins, we think the following algorithm could be considered:
> > >
> > > **Step 1:** select the first of $k$ nodes according to BFS.\
> > > **Step 2:** search and select the $k-1$ unassigned nodes that are the closest (in terms of hop distance) to the first selected node (e.g. with a neighborhood search)\
> > > **Step 3:** generate the spin values selected in Step 1. and Step 2. with Subgraph Tokenization\
> > > **Step 4:** repeat Step 1. until all nodes have assigned spin values

---

> > > > ### Comment · Reviewer_fy9s · 2023-08-21
> > > >
> > > > This alternative algorithm for node selection sounds interesting! I'm excited to see if it makes its way into a future paper.
> > > > My other concerns have been properly addressed, thank you.

---

### Official Review · Reviewer_ojQm · 2023-07-14

**Soundness:** 4 excellent
**Presentation:** 4 excellent
**Contribution:** 4 excellent
**Rating:** 8
**Confidence:** 4

**Summary:**

This submission proposes a novel unsupervised framework for solving graph combinatorial optimization problems. The proposed method is coined as VAG-CO, which autoregressively generates solutions to CO problems via annealing / reinforcement learning. Corresponding theoretical analysis is provided and numerical demonstration on various datasets are conduct.

**Strengths:**

- This paper is well written, and the technique is solid.
- The proposed autoregressive generation is appropriate for solving graph CO problems whose solutions are a set over the nodes, which could be represented with a binary vector.
- The authors use RL to optimize and carefully design an MDP to achieve that.
- An open sourced implementation is also provided.
- Solid technical analysis is provided.
- Various experiments on both simulated and realistic benchmark and on both MIS and MVC tasks.

**Weaknesses:**

This submission is great. A few issues are listed as follows.
- It seems the method is also applicable for other CO problems such as Minimum Dominate Set. Would it possible for the authors to elaborate about at least the possibility?
- It would be also good to evaluate under some subset of the MIS benchmark (https://github.com/maxiboether/mis-benchmark-framework).

**Questions:**

See above.

**Limitations:**

See above.

---

> ### Author Rebuttal · Authors · 2023-08-08
>
> We thank the reviewer for the thoughtful and constructive review. In particular, we are glad that the reviewer appreciates among other things our theoretical analysis, that the paper is "well written", and most importantly, that our method is "solid".
>
> **Applicability to other CO problem types, like Minimum Dominate Set (MDS):**
> Our method is designed to be directly applicable to problems that can be written in the form of an Ising Hamiltonian (Eq. 1). This class of problems is very broad and encompasses all of Karp’s famous 21 NP-complete problems (Lucas [2014]).
> The reviewer asked specifically about the MDS problem. This problem can indeed be formulated as an Ising-type CO problem. This is possible by using the reduction from MDS to the Set Cover problem (Kann [1992]). The latter can then be expressed as an Ising-type problem as shown in Section 5.1 in Lucas [2014].
>
> **Benchmarking on MIS-Benchmark:**
> We agree that using the MIS-Benchmark would be interesting. Nevertheless, we decided to focus on benchmarks from the most recent publications in the field, like Karalias et al. [2022] and Wang et al. [2023]. We added 3 new problem settings on 2 new CO problem types (Max-Cut and Max-Clique) works. For Max-Cut we significantly outperform the very recent work by Zhang et al. [2023] on the Barabási & Albert (BA) dataset with 200 - 300 nodes (BA 200 - 300). In case of the Max-Clique problem we benchmark on the ENZYMES and IMDB-Binary dataset and significantly improve upon the corresponding results in Karalias et al. [2022]. The results are shown in Tab. 4 and 5 in the rebuttal file.
>
>
> Kann [1992], “On the approximability of NP-complete optimization problems”, Doctoral dissertation, Royal Institute of Technology
> Lucas [2014], “Ising formulations of many NP problems”, arXiv:1302.5843
> Karalias et al. [2022], “Neural Set Function Extensions: Learning with Discrete Functions in High Dimensions”, arXiv:2208.04055
> Zhang et al. [2023], “Let the Flows Tell: Solving Graph Combinatorial Optimization Problems with GFlowNets”, arXiv:2305.17010
> Wang et al. [2023], “Unsupervised Learning for Combinatorial Optimization Needs Meta-Learning”, arXiv:2301.03116

---

> > ### Comment · Reviewer_ojQm · 2023-08-18
> >
> > Thank you for the feedback. I will keep my previous score.

---

### Author Rebuttal · Authors · 2023-08-08

We would like to thank the reviewers for their helpful reviews. Based on these reviews we could clarify several aspects of our work. In the following we will highlight the points of criticism that led to an extension of the presentation of our results and one point where we have difficulties in comprehending a fundamental point of criticism.

**Addition of runtimes**
Reviewer **rhwA** rightfully argued that the work would be improved if we added runtimes for our experiments. We did so for all of our experiments and whenever this information was available for reported results. All runtimes are now provided in the updated figures and tables in the rebuttal file. The obtained runtimes show that our autoregressive VAG-CO exhibits similar runtimes as recent mean-field (MF) methods. The reason for this surprising result is that the MF methods rely on conditional expectation which is a time consuming procedure. Importantly, our autoregressive method would not be able to achieve such remarkable runtimes if we would not employ our newly introduced Subgraph Tokenization (ST) technique. As shown in Fig. 3 in the rebuttal file an increased $k$ for ST yields better results and substantially reduced runtimes. We thank reviewer **rhwA** for raising this question since it highlights the benefit of ST.

**Experimental evaluation**
We evaluate our method on 8 real-world datasets and two synthetic datasets for in total 14 different hardness settings. We report results for 13 different methods. Therefore, we are somewhat surprised that the extend of the experimental evaluation is criticized by reviewer **59Wh** (both weaknesses are related to this aspect) and reviewer **rhwA** (weakness 4: "The experiments seem relatively limited in that comparison is only for two problem types.").
However, the reviewers are right in stating that our experiments focused on two CO problem types and that there are many others that would be interesting. Despite the very limited rebuttal period we, therefore, did our best to fully convince reviewer **59Wh** and reviewer **rhwA** of our experimental evaluation and can now report results on two additional CO problem types: Max-Cut and Max-Clique (Tab. 4 and 5 in the rebuttal file).
For Max-Cut we demonstrate that VAG-CO outperforms the very recent work Zhang et al. [2023] on their version of the Barabási & Albert dataset with 200-300 nodes (BA 200- 300). Similarly, our new results for Max-Clique on the ENZYMES and IMDB-Binary datasets improve upon the results by Karalias et al. [2022]. In both cases VAG-CO represents the state-of-the-art. We include runtimes for all of these results and report the Gurobi performance with various runtimes.
These additional results underpin the strong performance of VAG-CO and complement the numerous results that we already reported on MIS and MVC. In total, we are convinced that the updated experimental evaluation is extremely extensive in comparison to the standards in this field and that we could clearly establish the strong performance of VAG-CO.

In summary, VAG-CO is among the best performing methods on 10/11 real-world problems and the single best method on 8/10 real-world problems when no non-learned algorithmic components like conditional expectation are used. Since these real-world datasets that are frequently used in the recent literature are almost optimally solved we also include synthetic datasets that are known to be hard. VAG-CO outperforms the best learned methods on the real-world dataset on all 13 hard synthetic problem settings by a large margin.

**Unclear contribution and novelty**
While three reviewers agree that this work represents a valuable contribution the remaining reviewer **rhwA** writes "I do not understand the contribution of this paper." and "In general, I'm a little confused about what the overall contribution of this paper is in the context of the literature.".  In the following we would like to discuss several aspects of this point and hope that we can resolve potential misunderstandings.
Reviewer **rhwA** correctly pointed out several aspects of our work that are not new. As we argue below in our response to weakness (5) by reviewer **rhwA** this is correct but we never claimed these aspects to be new. For example, it is argued by the reviewer that the concept of annealing was already utilized in Sun et al. [2022] which  correct and also stated in L249. Then the reviewer calls into question the novelty of ST by writing that it “may be novel”. To the best of our knowledge it is novel and as our results show it is essential for the performance and runtime of VAG-CO. We hope that the reviewer will either outline the reasons for the doubt on this novelty or acknowledge it as an important new contribution.
We would like to point out that our main contributions:

(i) identification of a central limitation related to MF methods in numerous recent works,
(ii) introduction of Subgraph Tokenization to enable efficient autoregressive graph generation,
(iii) state-of-the-art results on several popular CO problems on real-world and on synthetic datasets,
(iv) a motivation for annealing from statistical learning theory,

are explicitly stated at multiple prominent places in the manuscript, including the abstract, the last paragraph of the introduction, and the conclusion. We hope that this clarification will facilitate the assessment of our contributions.

Finally, we are convinced that all other points raised in the reviews were clarified in our rebuttals below and we are happy to address any questions in the discussion period.

Zhang et al. [2023], “Let the Flows Tell: Solving Graph Combinatorial Optimization Problems with GFlowNets”, arXiv:2305.17010
Karalias et al. [2022], “Neural Set Function Extensions: Learning with Discrete Functions in High Dimensions”, arXiv:2208.04055
Sun et al. [2022], “Annealed Training for Combinatorial Optimization on Graphs”, arXiv:2207.11542

---

> ### Comment · Reviewer_fy9s · 2023-08-11
>
> Congratulations to the authors for the breadth of their numerical experiments, which are more than satisfactory. Indeed, the method as a whole may not be groundbreaking, but the authors are honest about it, and several key components appear quite novel. Besides, the outstanding performance on benchmark datasets deserves recognition. I'm thus raising my review score to 6.

---

### Decision · Program_Chairs · 2023-09-21

**Decision:**

Accept (poster)

**Comment:**

The paper has found a majority of positive reviews, and nice formulation, some novel training tricks, significant but some conceptual limitations (processing order ...), solid technical method and analysis, good experiments were noted. On the downside, rhwA raised a number of concerns, which could be addressed to a significant degree. Overall, the AC follows the majority view of the paper.
The exhaustive rebuttal has been positively acknowledged.